


# Machine learning-constrained projection of bivariate hydrological
# drought magnitudes and socioeconomic risks
Rutong Liu[1], Jiabo Yin[1*], Louise Slater[2], Shengyu Kang[1], Yuanhang Yang[1], Pan Liu[1], Jiali Guo[3,4],
Xihui Gu[5], Aliaksandr Volchak[6]
[1]State Key Laboratory of Water Resources Engineering and Management, Wuhan University, Wuhan,
Hubei, 430072, P.R. China
[2]School of Geography and the Environment, University of Oxford, Oxford, UK
[3]Engineering Research Center of Eco-environment in Three Gorges Reservoir Region, Ministry of
Education, China Three Gorges University, Yichang, Hubei 443002, China
[4]College of Hydraulic and Environmental Engineering, China Three Gorges University, Yichang,
Hubei 443002, China
[5]School of Environmental Studies, China University of Geosciences, Wuhan 430074, China
[6]Engineering Systems and Ecology Faculty, Brest State Technical University, Moskovskaya 267,
224017 Brest, Belarus
*Correspondence: Jiabo Yin (jboyn@whu.edu.cn)



**Abstract**
Climate change accelerates the water cycle and alters the spatiotemporal distribution of hydrological
variables, thus complicating the projection of future streamflow and hydrological droughts. Although
machine learning is increasingly employed for hydrological simulations, few studies have used it to project
hydrological droughts, not to mention the bivariate risks of drought duration and severity as well as their
socioeconomic effects under climate change. We develop a cascade modeling chain to project future bivariate
hydrological drought characteristics in 179 catchments over China, using 5 bias-corrected GCM outputs
under three shared socioeconomic pathways, five hydrological models and a deep learning model. We
quantify the contribution of various meteorological variables to daily streamflow by using a random forest
model, then employ terrestrial water storage anomalies and a standardized runoff index to evaluate recent
changes in hydrologic drought. Subsequently, we construct a bivariate framework to jointly model drought
duration and severity by using Copula functions and the most likely realization method. Finally, we use this
framework to project future risks of hydrological droughts as well as associated exposure of gross domestic
product and population. Results show that our hybrid hydrological-deep learning model achieves >0.8 Kling-
Gupta efficiency in 161 out of 179 catchments. By the late 21st century, bivariate drought risk is projected to
double over 60% catchments, mainly located in Southwest China. Our hybrid model also projects substantial
GDP and population exposures by increasing bivariate drought risks, suggesting an urgent need to design
climate mitigation strategies towards a sustainable development pathway.



# 1 Introduction

In a warming world, the acceleration of the global water cycle is expected to alter the regional and seasonal distribution of key hydrological variables such as precipitation and evapotranspiration (Allan et al., 2020). As precipitation patterns are particularly sensitive to changes in atmospheric forcing and local conditions, precipitation extremes are generally increasing globally, exacerbating spatial heterogeneity of precipitation (Donat et al., 2016; Tabari, 2020). A suite of Shared Socioeconomic Pathways (SSPs) has been proposed to simulate different possible future scenarios of societal responses to climate change, and these are employed to investigate the possible effects of long-term climate change (Meinshausen et al., 2020; Zhang et al., 2021). By using the SSP framework, numerous works have indicated that the redistribution of precipitation may lead to the decline of water storage in some regions, and intensify water scarcity in arid regions (Sönmez and Kale, 2018; Woolway et al., 2020; Yao et al., 2023). Under increasing atmospheric greenhouse gases, numerous studies have reported a widespread increase in drought events, even in areas with increasing annual runoff (Dai et al., 2018). The uneven distribution of precipitation and other meteorological elements under climate change complicates predictions of future runoff and drought.

China's socioeconomic development, and particularly its agricultural sector, is threatened by the rapid intensification of extreme hazards under climate change (Piao et al., 2010). Over the past years, China has been hit by severe drought events which have caused considerable damage to ecosystem productivity and socio-economic growth (Zhai and Zou, 2005; Yin et al., 2023). Water shortages, agricultural production, and associated ecological degradation are key challenges hindering the sustainable development of the North China Plain (Chen and Yang, 2013). Over the period of 1985-2014, drought accounted for about 19% of economic losses among all meteorological hazards (Chen and Sun, 2019). With continuing global warming, the economic losses from severe drought events might increase by over ten billions of US dollars per year by the late 21st century (Su et al., 2018). For instance, one extreme drought in Sichuan Province in 2022 resulted in power shortages and led to economic losses of 669 million dollars, underscoring the importance of projecting future droughts over China (Lu et al., 2023).

Droughts can be triggered by divergent mechanisms, and are thus distinguished according to the type of drought, such as meteorological and hydrological drought (Yihdego et al., 2019). The majority of studies have focused on meteorological droughts, which can then be translated to a hydrological drought, while fewer works have focused on hydrological drought probably due to lack of measurements like the standardized runoff index (SRI) (Barker et al., 2016; Kumar et al., 2016; Tirivarombo et al., 2018). Furthermore, hydrological droughts are not only affected by the water cycle but also by human interventions, which makes them difficult to accurately be predicted (Wu et al., 2021). Currently, the majority of drought impact assessments focus on the investigation of individual drought variables (i.e., drought duration, severity and intensity, etc.) through univariate probabilistic models and stochastic theory (Myronidis et al., 2018; Byakatonda et al., 2018; Zhang et al., 2022). However, univariate drought analysis cannot accurately describe



the probability of drought events, because droughts of either long duration or severe intensity can lead to
substantial socio-ecosystem damages (Castle et al., 2014; Udall and Overpeck, 2017). Therefore, the bivariate
framework based on Copula functions has been developed for drought projection, compensating for the
incompleteness of a single variable analysis (Ayantobo et al., 2017; Nabaei et al., 2019). At present, studies
on hydrological drought within a bivariate framework are still lacking. Beyond the choice of approach
(univariate or bivariate), the Gravity Recovery and Climate Experiment (GRACE) and GRACE-FO (GRACE
Follow-On) satellites now provide two decades of large-scale terrestrial water storage (TWS) data, which
captures the water deficit in various forms on land and can be used to monitor droughts (Schmidt et al., 2006).
The drought severity index based on TWS (TWS-DSI) can be used to monitor past drought events, which
also shows potential advantages in drought warning, forecasting, and projection (Nie et al., 2018; Pokhrel et
al., 2021).
In recent decades, many studies have used bias-corrected outputs from Global Climate Models (GCMs)
to project future hydrological drought scenarios (e.g., (Ashrafi et al., 2020; Kim et al., 2021; Dixit et al.,
2022). The growing application of machine learning has revealed high potential for improving the accuracy
of hydrological simulation and prediction (Mokhtar et al., 2021). In recent years, many machine learning
algorithms have been adopted in drought simulation and produce a good performance, such as wavelet neural
networks (WNNs) (Xiujia et al., 2022),  support vector machines (SVMs) (Zhu et al., 2021) and long short-
term memory neural networks (LSTMs) (Dikshit et al., 2021a)). These algorithms can be used to simulate
the evolution of future droughts and construct risk maps for drought contingency planning (Rahmati et al.,
2020). Among the different models, the LSTMs can effectively simulate short-term and long-term streamflow
series, and their performances have been validated at short temporal scales (Dikshit et al., 2021b; Kang et al.,

2023).

In this study, we project changes in bivariate hydrological drought characteristics (duration and severity)
and their associated socioeconomic risks under three SSPs (i.e., SSP1-26, SSP3-70, and SSP5-85) over 179
catchments in China. To achieve this, we combine five hydrological models and a deep learning model (i.e.,
the LSTM), and then drive the hybrid model with the five bias-corrected GCMs outputs under Coupled Model
Intercomparison Project phase six (CMIP6). Then, we employ a machine learning-based framework (i.e.,
Random Forest, RF model) to quantify the sensitivity of different meteorological variables to daily
streamflow. We employ the run theory and two drought metrics, the SRI and TWS-DSI, to identify and
explore recent changes in drought characteristics. In addition, we use Copula functions to build the bivariate
model of drought duration and severity during both reference and future periods. After identifying shifts in
bivariate drought characteristics based on the most likely realization approach, we project the exposure of
gross domestic product (GDP) and population to increasing drought risks in the future. Finally, we decompose
the uncertainties arising from different sources by employing the multivariate analysis of variance
(MANOVA) method. The paper provides a clear description of materials and methods used to analyze, and
then shows the difference between two drought indexes to assess drought conditions, the contribution of
meteorological factors to simulate streamflow, the validation of the accuracy of HTMs, the evolution of





univariate and bivariate droughts in future scenarios and the socioeconomic exposure to bivariate droughts.
We also make a discussion of uncertainty from multisource data and cascade model chain, and reflect on
limitations that could be improved to enhance the further study. All findings are summarized and targeted to
propose drought mitigation strategies.
**2. Methodology**

The workflow of this study is divided into four modules (Figure 1), described briefly below and detailed

in the following sections. In step 1, the hydrological models and LSTM are trained using the ERA5-Land
dataset, then the output of HMs is used as input to feed the LSTM, thus we build the hybrid terrestrial models
(HTMs). In step 2, the trained HTMs are validated using in situ streamflow observations, then driven by
using the outputs of five GCMs from the CMIP6 to project streamflow and the SRI series. In step 3, monthly
drought characteristics (i.e., drought duration and severity) are defined using run theory and combined with
Copula functions to construct a bivariate drought framework. Future bivariate drought change is evaluated
using the most likely realization method. Meanwhile, the TWS measurements from GRACE missions are
also employed to characterize recent changes in TWS-based droughts, which are also compared with the
hydrological droughts. In step 4, we employ future scenarios of GDP and population alongside our future
drought projections to produce a socioeconomic assessment of drought exposure over China. Finally, we
examine the contribution of uncertainty from different sources in projecting drought change and exposure.

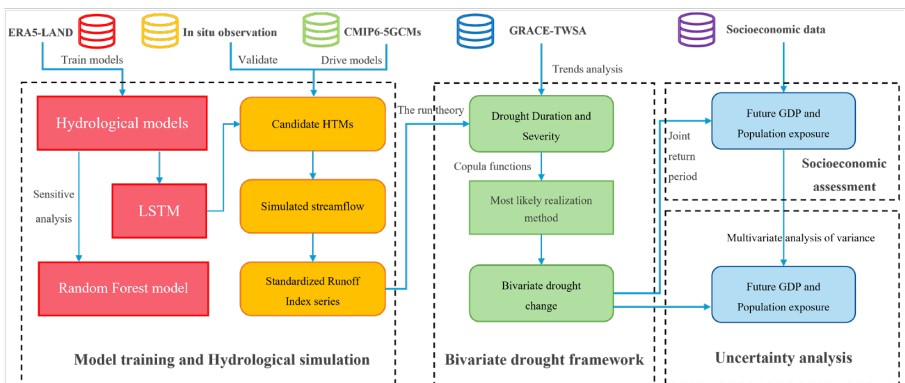


**Figure 1. Schematic flowchart of the method, including ML-constrained hydrological simulations, evaluation of**
**bivariate hydrologic drought characteristics and change, and the socioeconomic evaluation to drought exposure**
**under climate change.**
**2.1 Derivation of 2-meter relative and specific humidity**

The Clausius–Clapeyron relationship is used to derive saturated vapor pressure ($e_s$) and air temperature

($T$), and is expressed as follows (Koutsoyiannis, 2012):





$$e_s(T) = e_0 \exp\left[ \left(\frac{1}{T_0} - \frac{1}{T}\right)\frac{L_0}{R_0} \right]$$
(1)

where $T_0$, $e_0$, $L_0$ and $R_0$ are constants, with a value of 273.16 K, 611 Pa, 2.5×10⁶ J kg⁻¹, 461 J kg⁻¹ K⁻¹,
respectively;
Since near-surface relative humidity (*RH*) can't be directly obtained from the ERA5-Land dataset, the
2m temperature ($T_{2m}$) and dew-point temperature ($T_d$) are substituted into equation (1) to calculate *RH*:
$$RH = \frac{e_s(T_d)}{e_s(T_{2m})} = \exp\left[ \left(\frac{1}{T_{2m}} - \frac{1}{T_d}\right)\frac{L_0}{R_0} \right]$$
(2)

Then, the near-surface air pressure (*ps*) and $T_d$ are used to deduce the specific humidity (*SH*), which is
mathematically expressed as follows (Simmons et al., 1999):
$$SH = \frac{0.622 \times e_s(T_d)}{ps - 0.378 e_s(T_d)}$$
(3)

### 2.2 Sensitivity analysis on meteorological variables for runoff

The RF model is used to calculate the sensitivity to different meteorological variables for runoff,
including precipitation (*pr*), air pressure (*ps*), surface downwelling shortwave and longwave radiation ( *srsds*
*and srlds*), *RH*, *SH*, average temperature, maximum and minimum temperature. The contribution of a key
variable is derived by using the pre-established model, the perturbed meteorological variable and remaining
(non-perturbed) variables (Antoniadis et al., 2021; Green et al., 2020). The percentage change in streamflow
is derived from the following equation:
$$S_i = \frac{\text{mean}\left(R_{(i+1SD)} - R_{(all)}\right)}{\text{stdev}\left(R_{obs}\right)} \times 100\%$$
(4)

where $S_i$ indicates the sensitivity of streamflow to $i^{th}$ meteorological variable, which are *pr*, *ps*, *SH*, *RH*, *srlds*,
*srsds* and temperature; $R_{obs}$ is the observation of streamflow which has units of m³/s; $R_{(i+1SD)}$ is the simulated
streamflow by perturbing *i* by +1 SD; $R_{(all)}$ is the streamflow simulated by all meteorological variables; stdev
($R_{obs}$) represents the standard deviation of $R_{obs.}$

### 2.3 Deep learning-constrained hydrological modeling

### 2.3.1 Conceptual hydrological models

For preliminary hydrological simulations, we select five hydrological models to represent hydrological
characteristics under different environments. The GR4J (Génie Rural à 4 paramètres Journalier ) is a lumped
model with 4 parameters developed by Perrin et al. (2003). GR4J consists of two water store modules (runoff
yielding and routing) and uses daily rainfall and evapotranspiration as inputs to simulate streamflow series



(Kunnath-Poovakka and Eldho, 2019). This model has been successfully used to simulate hybrid runoff
processes in many continents (Shin and Kim, 2021; Gu et al., 2023). Additionaly, we use the temperature-
based method (Oudin et al., 2005) to estimate the potential evapotranspiration of the GR4J model.

The HBV (Hydrologiska Byråns Vattenbalansavdelning ) model was initially developed by the Swedish
Meteorological and Hydrological Institute for hydrological forecasting (BERGSTRÖM and FORSMAN,
1973). This model including five modules and one transform function to quantify hydrological variables (i.e.,
precipitation, snow, soil moisture, runoff, baseflow) (Bergström, 1995). It has been widely employed to
simulate streamflow, and it particularly has good capacity in simulating snowmelt runoff (Kriauciuniene et
al., 2013).

The HMETS (hydrological model of École de technologie supérieure) model contains 21 parameters
and two reservoirs (i.e., the saturated and vadose zones), which makes it simplified and efficient to complete
hydrological simulation (Martel et al., 2017).  The model can simulate six processes in water cycle, including
the accumulation, melst and refreezing of snow, water infiltration and routing, evapotranspiration (Qi et al.,
2020). It has been growly used for streamflow simulation under climate change and has shown well
performance (Chen et al., 2018).

The SIMHYD (simple lumped conceptual daily rainfall-runoff ) model is a daily rainfall-runoff model
developed by Porter and McMahon (1975). There are four types of runoff from different sourses: impervious
areas, infiltration, interflow, and groundwater store (Chiew et al., 2002). Although the model was developed
earlier, it has shown good accuracy in simulating runoff over China (Yu and Zhu, 2015).

The XAJ (Xinanjiang) model is a hydrological model, which can usually achieves better performance
in humid and semi-humid areas than in arid areas (Zhao, 1992). It is composed of a three-layer
evapotranspiration module with four parameters and separates the runoff into four components (i.e., surface
water, groundwater, interflow water and flow routing) (Tian et al., 2013). To date, it is widely reported that
the XAJ model usually show the best accuracy in simulating hydrological conditions in China (Hu et al.,
2005).

We use the SCE-UA (Shuffled Complex Evolution) approach with maximizing the objective function
(i.e., Kling-Gupta efficiency) to optimize these models (Duan et al., 1992). The most complete 20-year
observation period is selected to calibrate the models in each watershed. To calibrate the hydrological models,
a cross-validation method developed by Arsenault et al. (2017) is used, which employs the odd years of data
to calibrate models, and the even years of data to validate.

### 2.3.2 Hybrid scheme of hydrological model and machine learning

Recurrent neural network (RNN) models have had considerable success in hydrological modeling (Cho
et al., 2014; Sherstinsky, 2020). However, when considering long input sequences, RNNs struggle to capture
the relationships between distant points due to a phenomenon known as "long-term dependencies" (Yu et al.,
2019). With the development of deep leaning, this problem can be successfully avoided by using LSTMs.
A LSTM cell includes input, output and forget gates. The input gate determines which new information





can be stored in the cell state, and the forget gate identifies which information will be discarded from the cell
state. The output gate controls what part of the cell state is selected as the output. The updated cell state is a
combination of the information retailed and the new information to be added. By using this architecture, the
LSTM can avoid the problem of gradient vanishing or explosion during backpropagation, especially when a
series is long (Gers et al., 2000). The LSTM can be expressed as follows:
$$fg_t = \sigma(W_{hf}hs_{t-1} + W_{xf}x_t + b_f) \tag{5}$$

$$ig_t = \sigma(W_{hi}hs_{t-1} + W_{xi}x_t + b_{fg}) \tag{6}$$

$$\tilde{c}_t = \tanh(W_{h\tilde{c}}hs_{t-1} + W_{x\tilde{c}}x_t + b_{\tilde{c}}) \tag{7}$$

$$c_t = fg_t \cdot c_{t-1} + ig_t \cdot \tilde{c_t} \tag{8}$$

$$og_t = \sigma(W_{oh}hs_{t-1} + W_{ox}x_t + b_o) \tag{9}$$

$$hs_t = og_t \odot \tanh(c_t) \tag{10}$$

where $x_t$, $fg_t$, $ig_t$ and $og_t$ are input variables, and forget, input and output gates at time $t$, respectively; $W_i$,
$W_{\tilde{c}}$, $W_f$ and $W_o$ are the weights of each gate; the operator ' $\odot$ ' is the symbol for the dot product of two vectors;
$c_t$ and $hs_t$ are the cell state of the LSTM and the hidden unit at the time $t$, $c_{t-1}$ and $hs_{t-1}$ at the former time
$t-1$; $\tilde{c_t}$ is the activation function of hidden layer; $b_i$, $b_f$, $b_o$ and $b_c$ are bias itemsand the; $\sigma(\cdot)$ and $\tanh(\cdot)$
are the sigmoid function and the hyperbolic tangent function, respectively; at the initial moment, cell and
hidden states are set to zero arrays.

The hydrological outputs together with other climate variables are used as inputs to feed the LSTM

model (i.e., the HMs are thus constrained by the LSTM). Because changes in meteorological variables require
some time to converge before they are reflected in the runoff, it is essential to calculate the lag time caused
by the flow convergence for the model. The catchment response lag time $d$ is defined as the time during
which precipitation accumulates in the river to generate runoff for the gauge downstream, and is
mathematically expressed as follows (Berne et al., 2004; Ganguli and Merz, 2019):
$$d = 2.51A_d^{0.4}[\text{ hrs }] = 0.11A_d^{0.4}[\text{ days }] \tag{11}$$

where $A_d$ (km$^2$) represents the catchment area; meteorological variables from day $T$-d to day $T$ are employed
to drive HTMs.

We combine the five hydrological models with LSTM to construct five HTMs. To compare the

performance of the HTMs, we use ten HTMs as candidates for streamflow simulation in each catchment. The
calibrated HTMs are then driven by the outputs of five GCMs under each SSP (aggregated to produce a basin
average series) during 1985-2100 over 179 catchments to project future daily streamflow.





**2.4 Drought indexes and run theory**

The TWS-DSI is employed to measure the degree of terrestrial drought severity (Zhao et al., 2017). It

is a dimensionless standardized water storage anomaly index, which can indicate terrestrial drought
conditions when below the mean standard value. The TWS-DSI can be mathematically expressed as follows:
$$TWS\text{-}DSI_{x,y} = (TWS_{x,y} - \overline{TWS_y}) / \sigma_y \tag{12}$$

where $TWS_{x,y}$ is the TWS at year $x$ and month $y$; $\overline{TWSA_y}$ and $\sigma_y$ represent the means and standard deviation
of TWS at month $y$, respectively.

The SRI is a measure of the variability of runoff for a given duration based on the percentage of

accumulated runoff. (Shukla and Wood, 2008). To calculate the SRI, we simulate the retrospective time series
of streamflow and fit the sample series to a probability distribution. The SRI is considered to follow a Pearson
type-III distribution (Vicente-Serrano et al., 2012), and is calculated as follows:
$$SRI = \begin{cases} -\left(r - \dfrac{c_0 + c_1 r + c_2 r^2}{1 + d_1 r + d_2 r^2 + d_3 r^3}\right) & 0 < F(x) \le 0.5 \\[3mm] r - \dfrac{c_0 + c_1 r + c_2 r^2}{1 + d_1 r + d_2 r^2 + d_3 r^3} & 0.5 < F(x) \le 1 \end{cases} \tag{13}$$

where $r = \sqrt{\ln\left[\dfrac{1}{F(x)^2}\right]}$ ; $F(x)$ is the cumulative probability density of SRI; $c_0$, $c_1$, $c_2$, $d_1$, $d_2$ and $d_3$ are
the empirical constants, taken as 2.516, 0.803, 0.010, 1.433, 0.189, 0.001, separately.

After calculating the two drought indexes, the degree of water deficit can be determined according to

the Grades of Meteorological Drought and the previous classification (Dikici, 2020). Table S1 presents the
drought classification and thresholds used for identifying drought degrees. The run theory is employed to
obtain characteristics of drought events from the time series (Yevjevich, 1967). When the drought index is
below the mild drought (i.e.,≤-0.5 drought index), a drought event is detected (Figure 2), and then the drought
duration and drought severity are extracted.

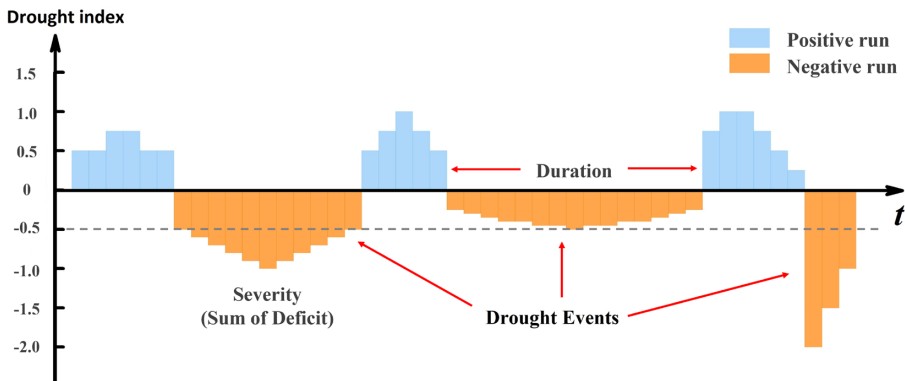


**Figure 2. Drought duration and severity identification based on run theory, where -0.5 denotes the drought threshold (grey dash line).**

**2.5 Socioeconomic exposure assessments based on the Copulas and most likely realization**

After extracting the drought duration (*D*) and severity (*S*), we fit their marginal distributions with seven distributions shown in Table S2. The OR case (i.e., a bivariate drought event is identified with either a high severity or long duration) of the joint return period (JRP) under a Copula-based framework is used to quantify the occurrence of drought events (Yin et al., 2020). The joint distribution of drought duration and severity is constructed by using a Copula function, which is valuable for describing correlated hydrological variables (Li, 1999). Unlike univariate drought frequency analysis, the JRP within a bivariate framework can be represented by an isoline, which contains infinite combinations of multivariate variables. It is important for risk assessments to select a representative combination along the isoline. Previous studies have typically selected joint design values according to the same frequency hypothesis, but this approach lacks a statistical basis and poorly describes the physical characteristics of droughts (Yin et al., 2018). In this paper, the joint probability density is used to optimize the most likely realization, which is mathematically expressed as follows:

$$
\begin{cases}
(d^*, s^*) = \arg\max f(d,s) = c[F_d, F_s] \cdot f_d \cdot f_s \\
C[F_d, F_s] = 1 - \mu / T_{or} \\
c[F_d, F_s] = \dfrac{dC(F_d, F_s)}{d(F_d)d(F_s)}
\end{cases}
\tag{14}
$$

where $c[F_d, F_s]$ is the Copula probability density function; $f_d$ and $f_s$ are the fitted probability density functions of *D* and *S*, respectively; $F_d$ and $F_s$ are the marginal distribution of *D* and *S*, respectively; $(d^*, s^*)$ is the most likely realization under a given JRP $T_{or}$; $\mu$ is the mean inter-arrival time between two consecutive droughts.



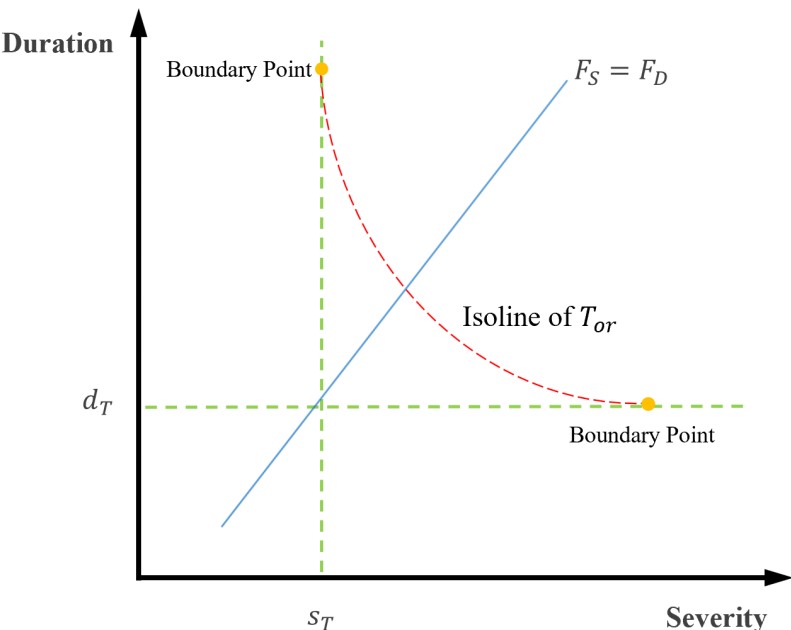

Figure 3. Joint distribution of drought duration and severity under a critical $T_{or}$. The green lines are two arbitrary values of duration and severity. The red line is the isoline line of two variables under a critical $T_{or}$, and the blue line denoted the traditional equal-frequency assumption.

Socioeconomic exposure has previously been defined as ranging from 0 to 100% in the future period (Gu et al., 2020a), but dynamically shifting climate risks cannot be represented under this static definition. Here, the socioeconomic exposure is defined by considering the shift in JRP, and is expressed at the catchment scale as follows:

$$E_{POP} = \frac{T_h I(T_h - T_f)}{T_f A_d} \times POP \quad (15)$$

$$E_{GDP} = \frac{T_h I(T_h - T_f)}{T_f A_d} \times GDP \quad (16)$$

where $E_{POP}$ and $E_{GDP}$ demote the population and GDP exposure; $T_h$ and $T_f$ demote the historical and future JRP, respectively; $I(\cdot)$ denotes the controlling function, which is 1 when $T_h - T_f > 0$, or 0 when $T_h - T_f \geq 0$ is recorded; *POP* (*GDP*) denotes the population (GDP) of a given catchment in the future climate.



**2.6 Quantifying the uncertainty contributed by different sources**
Uncertainties in the future drought projections can arise from the SSPs, GCMs and HTMs. During both
historical (1985-2014) and future periods (2071-2100), the combination of 3 SSPs, 5 GCMs and 5 HTMs
through the impact modeling chain resulted in 150 hybrid combinations. The overall uncertainty is calculated
from the variance of the future estimated JRP relative to the historical 50-year droughts. To partition the
uncertainty from different sources of data and their interactions effects, the MANOVA is used and expressed
as follows (Weinfurt, 1995):
$$\Delta y_{x,y,z} = M + S_x + G_y + H_z + I_{x,y,z} \tag{17}$$

where $M$ denotes the mean change of all indicator in models; $S_x$, $G_y$ and $H_z$ denote the impact on
indicators of the $x^{th}$ SSP, $y^{th}$ GCM and $z^{th}$ HTM, respectively; $I_{i,j,k}$ is the overall impact arising from the
interactions of different sources. And the overall variance $V$ is then expressed as follows:
$$V = VS + VG + VH + VI_{SG} + VI_{SH} + VI_{GH} + VI_{SGH} \tag{18}$$

where $VS$, $VG$, $VH$ are the variance from the SSPs, GCMs and HTMs, respectively. $VI_{SG}$, $VI_{SH}$, $VI_{GH}$
and $VI_{SGH}$ denote the variance caused by the coupling between different sources of data. The contribution
of each source to the overall uncertainty is quantified by the variance of each source by the total variance.
**3. Data and materials**
**3.1 In situ observation dataset**
We use a gridded meteorological dataset with 0.5° × 0.5° resolution, including daily temperature
(maximum, minimum and average, ℃) and daily precipitation (mm) from 1961 to 2018, provided by the
National Meteorological Bureau of China. The dataset is regarded as the latest gridded meteorological dataset
in China and has been applied to some studies (e.g., Wu et al., 2018; Yin et al., 2021a,b). Meanwhile, we
gathered the daily streamflow of 463 in situ hydrological stations spanning different periods during 1961-
2018. The hydrological stations are densely distributed in East China, while West China has a sparser
distribution.  Through rigorous data quality checks, 179 unnested basins with at least 20 years of data are
selected, covering nine major watersheds in China. For more details on streamflow data processing and
catchment screening, please refer to Yin et al. (2021b).
**3.2 GRACE/GRACE-FO measurements**
Temporal variations in the Earth's gravitational field observed by GRACE satellites have been used to
retrieve TWS data (Tapley et al., 2004). Many international institutes have released the TWS mascon products
at a monthly scale, including the JPL (Jet Propulsion Laboratory of the California Institute of Technology),



the GSFC (Goddard Space Flight Center of NASA), and the CSR (Center for Space Research of the
University of Texas). As these three mason solutions are produced different spatial resolutions, we produce
a blended TWS data based on the average of JPL, GSFC and CSR with 0.5°×0.5° resolution from 2002 to
2022, and fill the missing data using a linear interpolation approach (Yin et al., 2022).

**3.3 ERA5-Land dataset**

ERA5-Land is a dataset that consists of a large volume of meteorological variables, including
precipitation, temperature and air pressure etc. The spatial resolution of dataset is 9 km and the temporal
resolution is one hour (Yilmaz, 2023). Under the latest global reanalysis and the lapse rate correction, the
ERA5-Land reanalysis dataset provides a substitute for unavailable observed weather data, by taking the
effect of altitude on the spatial scheme of climate variables into consideration (Pelosi et al., 2020). Six
variables are used in the study (i.e., $pr$, $ps$, $T_{2m}$, $T_{dew}$, $srlds$, $srsds$) and aggregated to a daily scale from the
hourly scale before conducting data analysis.

**3.4 Bias-corrected GCM outputs and socioeconomic scenarios**

The climate outputs of five GCMs under historical scenario and three SSPs (i.e., SSP1-26, SSP3-70,
SSP5-85) under CMIP6 are used to represent climate scenarios. The series of bias-corrected variables have
been downscaled to 0.5° × 0.5°resolution from 1850 to 2100 under the Intersectoral Impact Model
Intercomparison Project 3b (ISIMIP3b) (Lange, 2019). To reduce the systematical biases of CMIP6 raw
outputs, seven variables from the bias-corrected ISMIP3b dataset have been used, namely temperature (daily
average, maximum and minimum), $pr$, $ps$, $srsds$, $srlds$, $RH$ and $SH$.
Population and GDP data under three SSPs are employed to evaluate the potential socioeconomic risks
of drought in a warming world. An open-access population dataset is adopted which takes into consideration
the universal two-child policy, the census results and the statistical annual report (Jiang et al., 2017). The
economic index from 2010 to 2100 is estimated based on the Cobb-Douglas and Population-Environment-
Development model (Jiang et al., 2018). All of the data have been previously used to assess the socio-
economic impact of extreme hydrologic hazards (Yin et al., 2022; Yin et al., 2023).

**4. Results**

**4.1 Observed changes in SRI and TWS based drought**

As there are insufficient streamflow observations to compute the SRI in northwest China, we also
employ the TWS-DSI as a supplement. This approach enriches the variety of water storage or flux being
evaluated. Trends in drought characteristics (i.e., frequency, duration and severity) are estimated by using the
GRACE/GRACE-FO dataset and observed runoff across China. Figure 4 and Figure 5 show the drought
trends based on the TWS-DSI and SRI, respectively. Overall, the two indexes show similar trends in most
catchments, suggesting that drought hazards have increased in recent decades. TWS-DSI droughts have





increased in 54% of areas, which are mainly located in the Qinghai-Tibet Plateau, the North China Plain and
the northwestern Xinjiang Province. Likewise, SRI droughts have increased over 51% of studied catchments,
which mainly dominates northeastern and southeastern China.  The severity of droughts measured by the
TWS-DSI index is twice of the hydrological drought, primarily because the TWS-DSI metric incorporates
all vertical water fluxes, offering a comprehensive view of shifts in water scarcity. Some locations exhibit
discrepancies depending on the index considered. For instance, droughts in the Qinghai-Tibet Plateau and
Northeast China show opposite trends. Anomalies in the Qinghai-Tibetan plateau may be explained by the
transformation of snowpack melt into surface runoff under the influence of climate change, which helps
compensate for the lack of surface water in the area (Stewart, 2009). The discrepancy observed in
Northeastern China could potentially be linked to the rise in soil moisture from increased infiltration, which
causes a higher proportion of water to be stored within the soil than at the surface, interfering with the
quantification of hydrological drought (Wang et al., 2017). Finally, both indicators show a consistent positive
drought trend in most areas of China and particularly the North China Plain and Pearl River Basin.



Figure 4. Trends in drought frequency, duration and severity based on the TWS-DSI from 2002 to 2022 using three GRACE/GRACE-FO products (a-i) and the blended data (j-l).



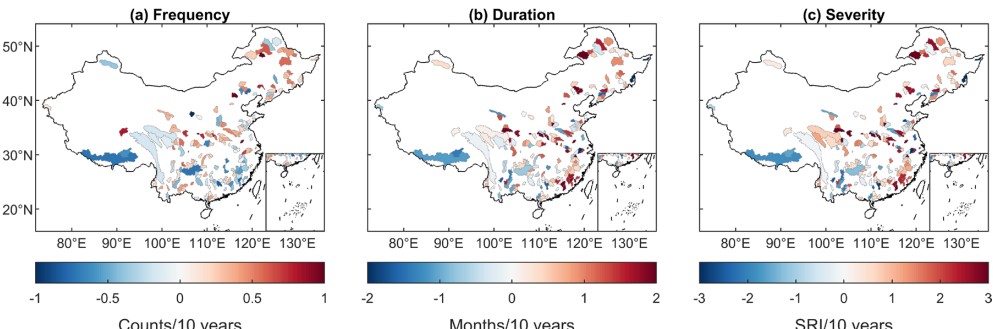

Figure 5. Trends in drought frequency, duration and severity based on SRI over China.

**4.2 Machine Learning-constrained streamflow simulation and model evaluation**

The RF model is used to quantify the sensitivity of streamflow to different meteorological variables (Figure 6). Precipitation typically plays a major role in generating runoff in Southeast China, although *SH* plays the most important role in some regions such as Central, Southwest and Northeast China. Over 30% and 38% of stations show a sensitivity rate of >10% in Western and Northeastern China, respectively. In contrast, *RH* and shortwave radiation have a negative contribution to streamflow; especially shortwave radiation, which has a pronounced negative sensitivity in 394 stations probably due to enhanced evapotranspiration (Ma et al., 2019). In general, *RH* contributes to increasing streamflow over most regions of China, but the opposite effect is observed in 179 stations mainly located in Southwestern China, Yellow River and Huaihe River basins. This is the result of the mutual feedback of water and heat dynamics (i.e., saturated vapor pressure increases with warming and intensifies evaporation, leading to a decrease in surface water), which was also found by Liu et al. (2017). The temperature has a positive contribution in Northeast China, suggesting that runoff in this region is likely to increase in the context of climate warming, leading to a reduction in drought over the regions.

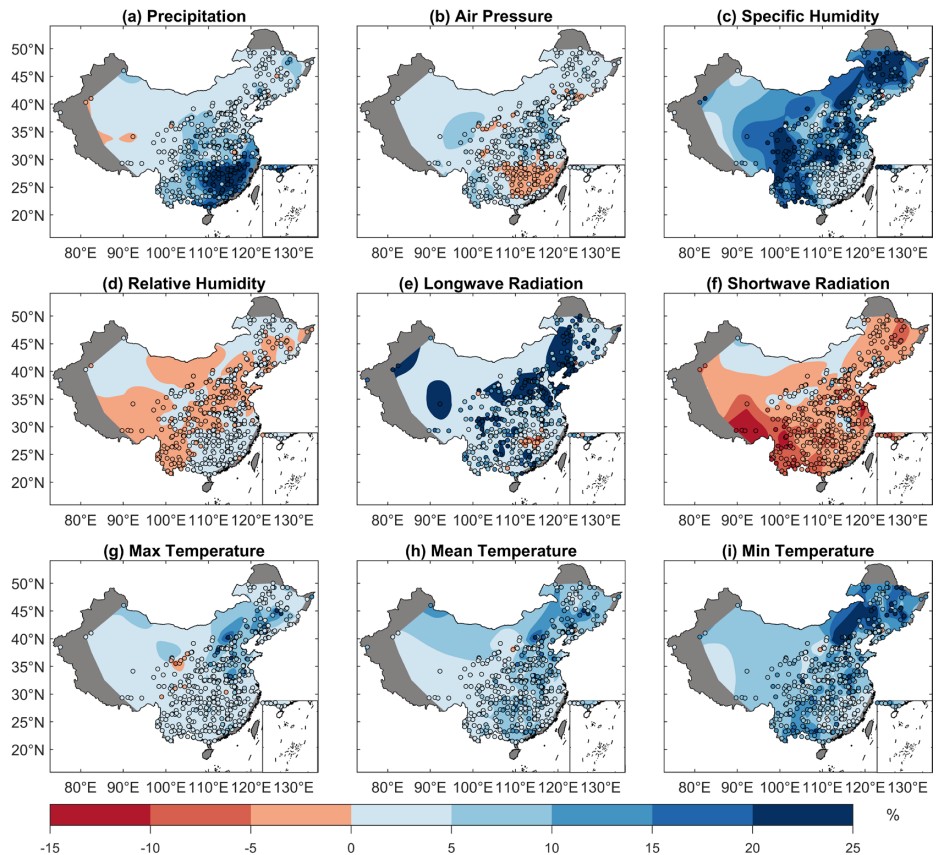

**Figure 6. Sensitivity of meteorological variables to daily streamflow. The figure uses a thin plate smoothing spline method to interpolate the point-based station data (circles). Gray areas indicate missing data.**

The performances of simulated streamflow by different HTMs are shown in Figure 7. The model that has the largest KGE is considered to be the best-performing in each catchment. In Fig 7. (a) and (b), the GR4J and GR4J-LSTM performed best in 77 out of 179 studied catchments. The median KGE value of GR4J is higher than 0.83, revealing a superior performance than the other hydrological models. Subsequently, the XAJ and XAJ-LSTM are the best models in 57 catchments, mainly located in the southern Yangtze River.. Last, the HBV and HBV-LSTM performed best in only 10 catchments, where the streamflow are impacted by snowfall in plateaus and northern frozen areas. All catchments exhibit KGE values greater than 0.9 during the calibration period in Figure 7c, showing good performance in simulation. During the validation period, only 18 catchments have KGE values below 0.6, and most of the catchments have KGE values greater than 0.8 in Figure 7d. In summary, the trained models simulate streamflow well in all the studied catchments. Additionally, the KGE values in the southern region are generally higher than those in the northern region during the validation period, which is consistent with previous hydrological simulation works (Gu et al., 2020b, 2021). This phenomenon may be attributed to the higher dependence of streamflow on rainfall in South China, which is governed by a humid climate pattern (Zheng et al., 2022).



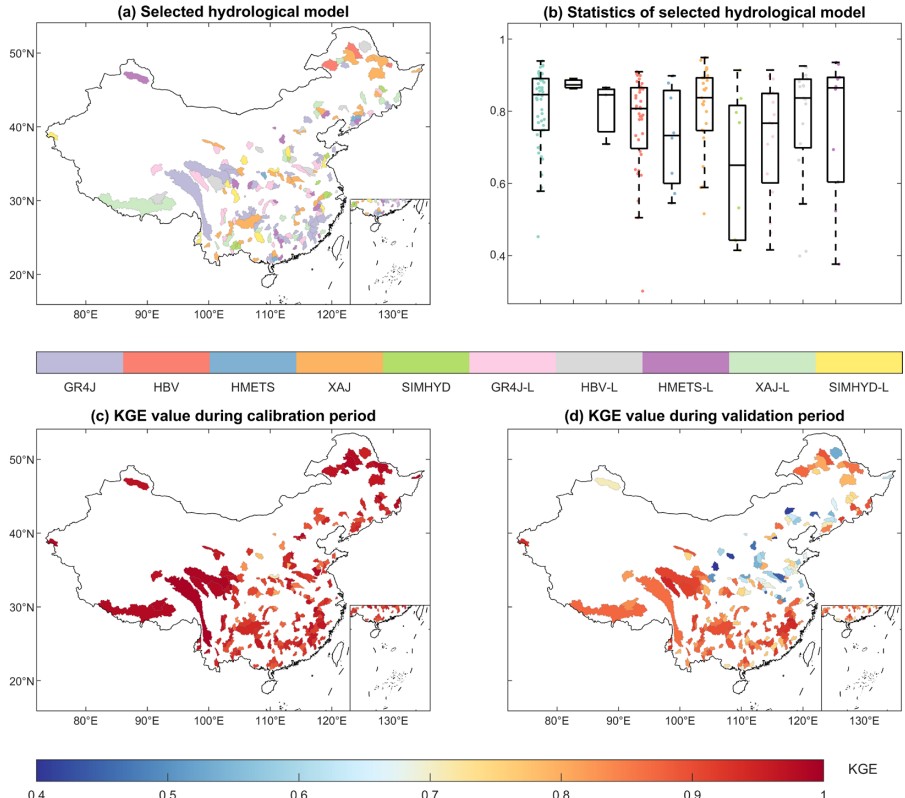

**Figure 7. Hydrological simulation performances of all candidate models. (a), The best-performing model with the highest KGE value. (b), Boxplots of all catchments for ten HTMs indicated by KGE values. (c)-(d), The highest KGE values during the calibration (c) and validation (d) period, respectively.**

### 4.3 Projected changes in univariate drought characteristics

We project the future daily runoff series by driving the HTMs with the bias-corrected CMIP6 variables, and then we estimate the monthly SRI to identify drought duration and severity. Based on the maximum Bayesian Information Criterion (BIC), we select the best-performing marginal distributions for duration and severity from seven candidate distributions, based on historical data for each catchment. Figure 8 and Figure 9 show the multi-model ensemble average severity and duration for the 50-year historical return period (RP).

In western China, we project a significantly increasing drought trend under the three SSPs, which indicates potential for increased water scarcity and more frequent extreme drought events. In Southeast China, we project that droughts are likely to intensify under SSP3-70 but not under SSP5-85. It is generally considered that SSP5-85 is accompanied by higher carbon emissions than that of SSP3-70 (O'Neill et al., 2016). However, future works also take significant action to control the extent of climate change combined with strong climate policies under SSP5-85 (Fujimori et al., 2017). As a result, there is no deterioration of drought severity with policy interventions, which emphasizes the significance of ensuring the implementation





of climate strategies. In northern China, in contrast, we find that future drought risks are projected to decrease
under the three scenarios, which is possibly related to more moisture convergence from the East Asian
monsoon circulation as the warming climate (Chowdary et al., 2019).

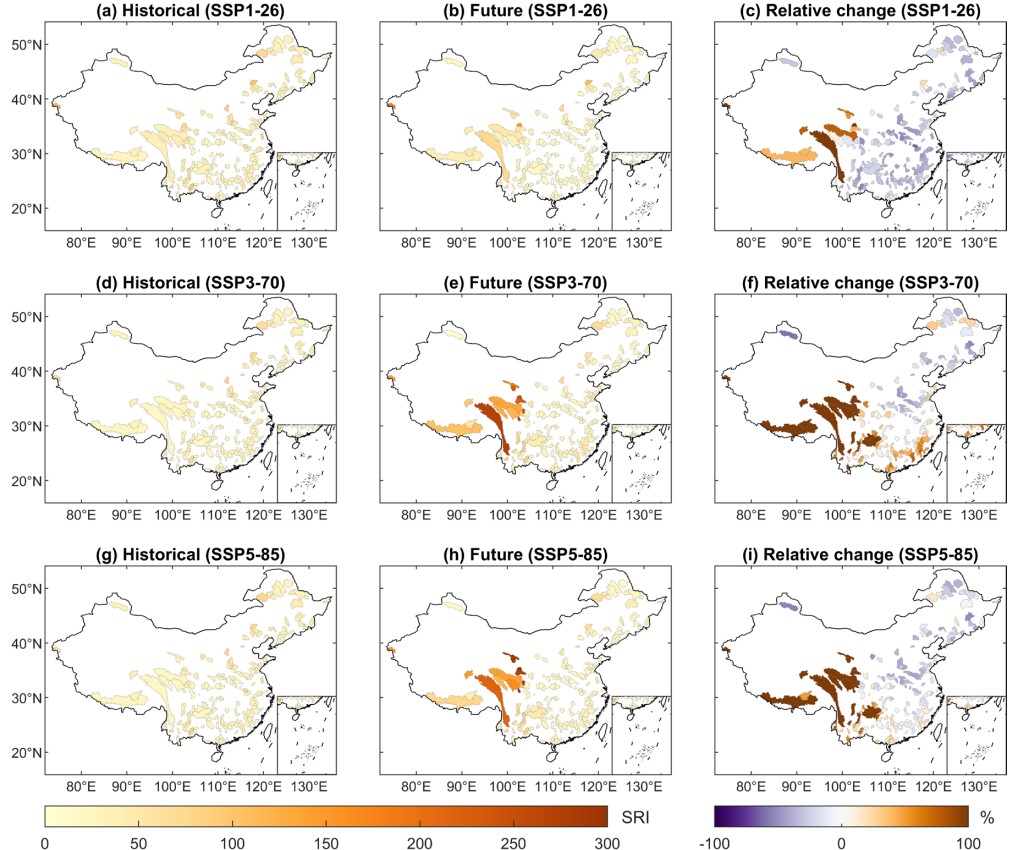

**Figure 8. Multi-model ensemble average design severity (dimensionless) under a 50-year RP for three SSPs, and**
**relative changes (%) in 2071-2100 compared to 1985-2014.**

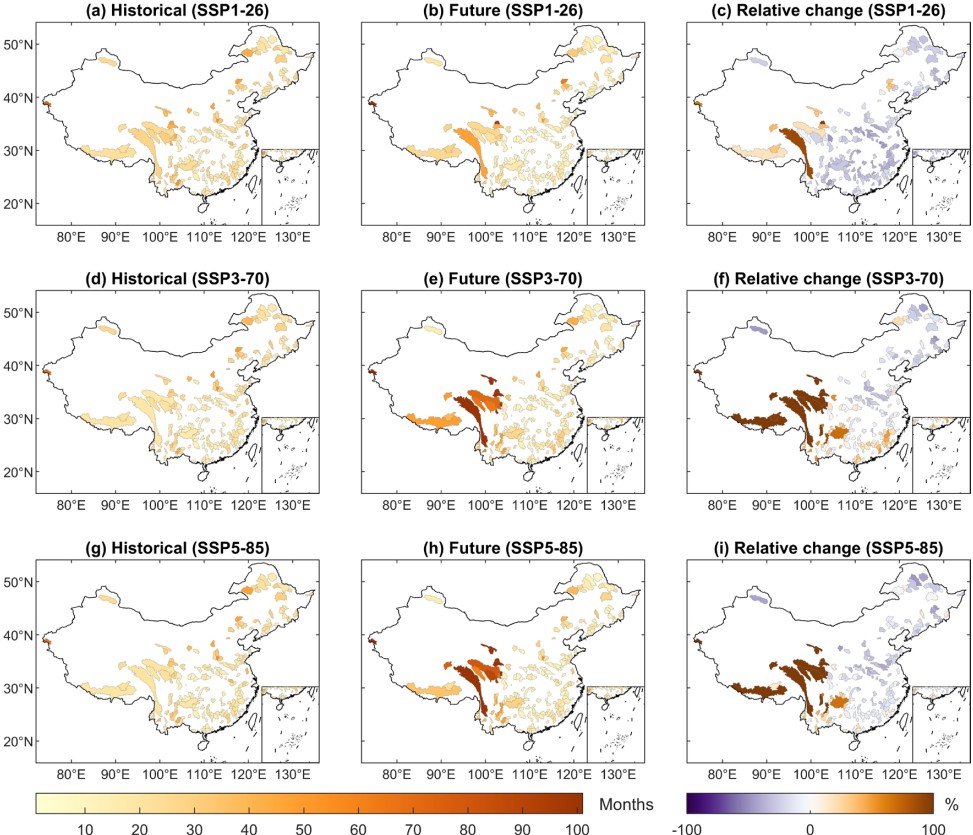

**Figure 9. Multi-model ensemble average design duration (months) of the multi-model for a 50-year RP for three SSPs, and relative changes (%) in 2071-2100 compared to 1985-2014.**

We display the relative change of drought characteristics under 50-year RP for all catchments for five GCMs under the three SSPs using violin plots (Figure 10). For most catchments, the relative change of drought duration and severity is negative. However, the relative change under some scenarios reached a maximum of 400%, highlighting the extreme change of drought. The median relative change of severity based on the IPSL_CM6A_LR under SSP3-70 are 30%, and 22% of catchments have a relative change over 200%, representing the most severe case of drought evolution. Furthermore, the distributions of the projections based on the MPI-ESM1-2-HR, MRI-ESM2-0 and UKESM1-0-LL models are highly skewed and bimodal under SSP3-70 and SSP5-85, revealing substantial spatial heterogeneity across China. Overall, the severity and duration of droughts slight increase in some catchments and have the risk of extreme intensification as global warming.



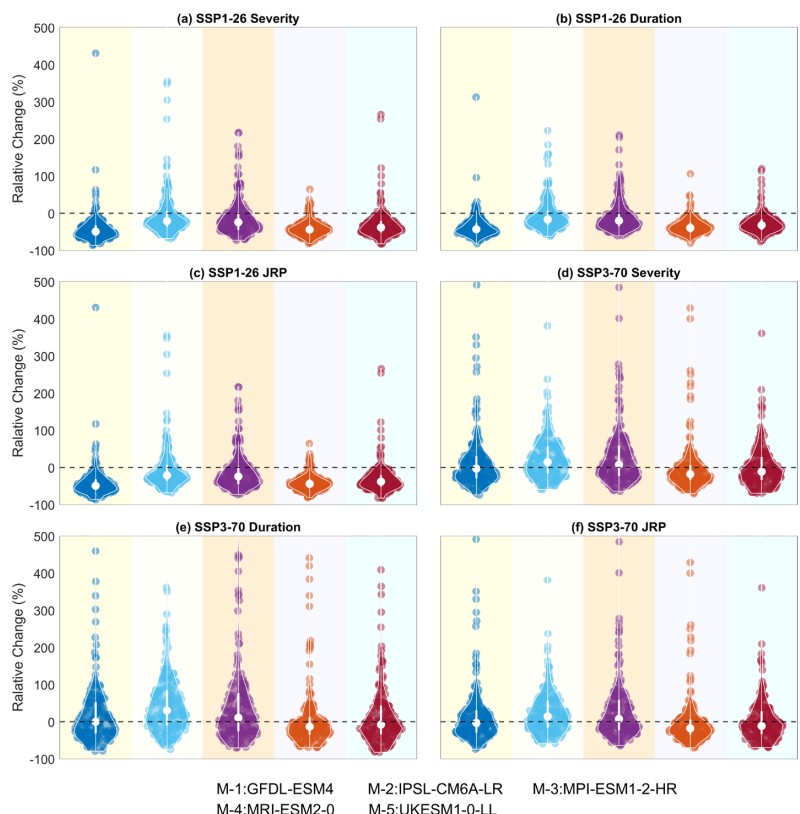


**Figure 10. Violin plots of relative changes (%) in severity and duration to the historical drought event with 50-year RP under three SSPs. The white circles are the median values of relative changes.**


### 4.4 Bivariate drought changes and corresponding socioeconomic risks

To capture the complex dependence structure between drought severity and duration, we use a Copula
function to quantify the bivariate risk of hydrological droughts under climate change. Changes in the JRP of
the historical (1985-2014) drought event with 50-year JRP in the future (2071-2100) period are shown in
Figure 11. The medians of the projected future JRP are 38.78, 14.52 and 19.24 under SSP1-26, SSP3-70 and
SSP5-85, respectively. For 69% and 60% catchments under SSP3-70 and SSP-5-85, we find the JRP of the
50-year drought is reduced to less than 25 years in the future period, suggesting that the risk of drought
increases over 2 times in these catchments. Besides, we find a marked increase in the number of catchments
with increased drought risk compared to the univariate drought assessments. The JRP of catchments in
Northeastern and Central China tends to decrease, suggesting higher changes in risks than univariate
assessments. This result is consistent with previous studies (He et al., 2011; Xu et al., 2015), which indicates
that the use of bivariate drought analysis can synthesize the effects of two drought characteristics.
Future GDP and population exposed to increasing bivariate drought risk under three scenarios are shown
in Figure 12. The eastern coastal regions have a higher significant economic exposure such as the Huaihe



River Basin, the Yangtze River Basin and the Pearl River Basin, which is consistent with the distribution of

economically developed regions in China. The medians of GDP exposure are 5.5, 9.8 and 14.3 million

dollars/km$^2$ under three SSPs respectively, which indicates the vulnerability of economic losses to drought

disasters under global warming. The population affected by drought is mainly located in the southern Yangtze

River Basin and the Huaihe River Basin under SSP3-70, as the median exposure is 525 and 205 people/km$^2$

under SSP3-70 and SSP5-85, respectively. This is because the increase in population is higher in the Sichuan,

Guangdong and Zhejiang provinces than in other Chinese provinces under SSP3-70 (Chen et al., 2020).

Overall, the exposure of GDP and population shows large heterogeneity in their sensitivity to different

scenarios, and the distribution of the affected catchments is consistent with economic and social development.

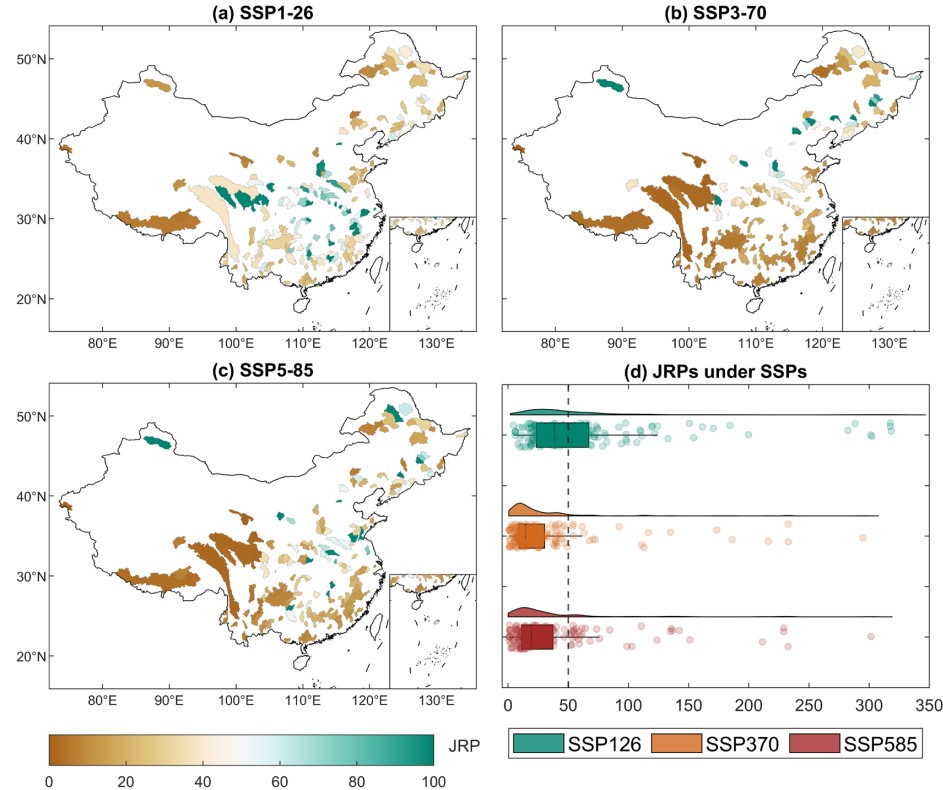

**Figure 11. The future multi-model ensemble means JRP of the historical drought with a 50-year RP based on the bivariate approach. The future JRPs of 179 catchments under three SSPs are presented in (a)-(c), while (d) displays raincloud plots of the projected JRP under each SSP.**

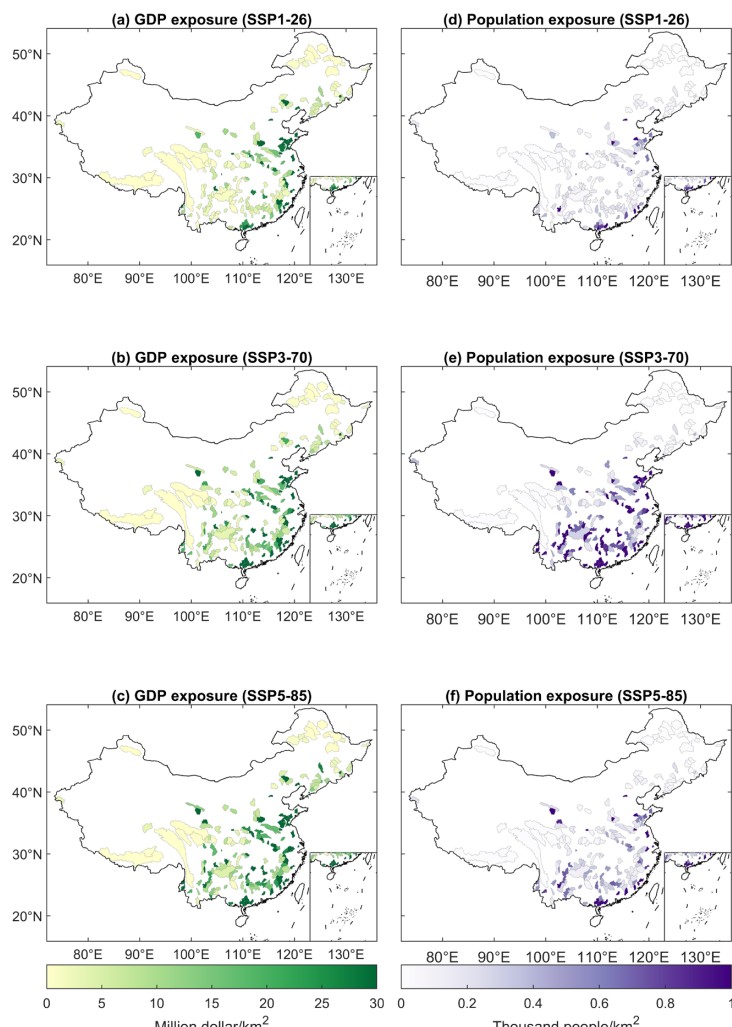

**Figure 12. The multi-model ensemble means exposure of GDP (a-c) and population (d-f) to bivariate drought characteristics under different SSPs in the future period.**

## 5. Discussion

### 5.1 Uncertainty decomposition

The overall uncertainty in our projections arises from the different SSPs, GCMs and HTMs as well as their interactions. We assemble these seven sources using MANOVA (Figure 13). For GDP and POP exposure, we find HTM is the main source of uncertainty, and contributes 27.55% and 26.14% uncertainty, respectively. This indicates that the quality of the HTM is important for the accuracy of socioeconomic predictions. Likewise, the GCM and GCM-HTM provide over 30% of the uncertainty in GDP and population exposures,

which indicates the critical importance of bias-corrected GCM outputs for accurate projections. Further, the

contributions of the SSPs to population exposure is 1.5 times than that of GDP exposure, which shows that

the effect of climate change is greater for POP exposure than GDP exposure. In particular, the independent

factors (i.e., SSP, GCM, HTM) contribute over 50% to the uncertainty of GDP and population exposures,

suggesting that GDP and population exposures are less responsive to complex coupling. In contrast, the

coupled factors (i.e., the combination of SSP, GCM or HTM) mainly contribute to the uncertainty of the JRP,

accounting for 82.63% of the overall uncertainty, especially the SSM-GCM-HTM, which accounts for 36.97%

of uncertainty. Finally, the relatively low contribution of the choice of SSP, SSP-GCM and SSP-HTM to JRP

uncertainty indicates that the future risk projection uncertainty is relatively stable in future risk projections

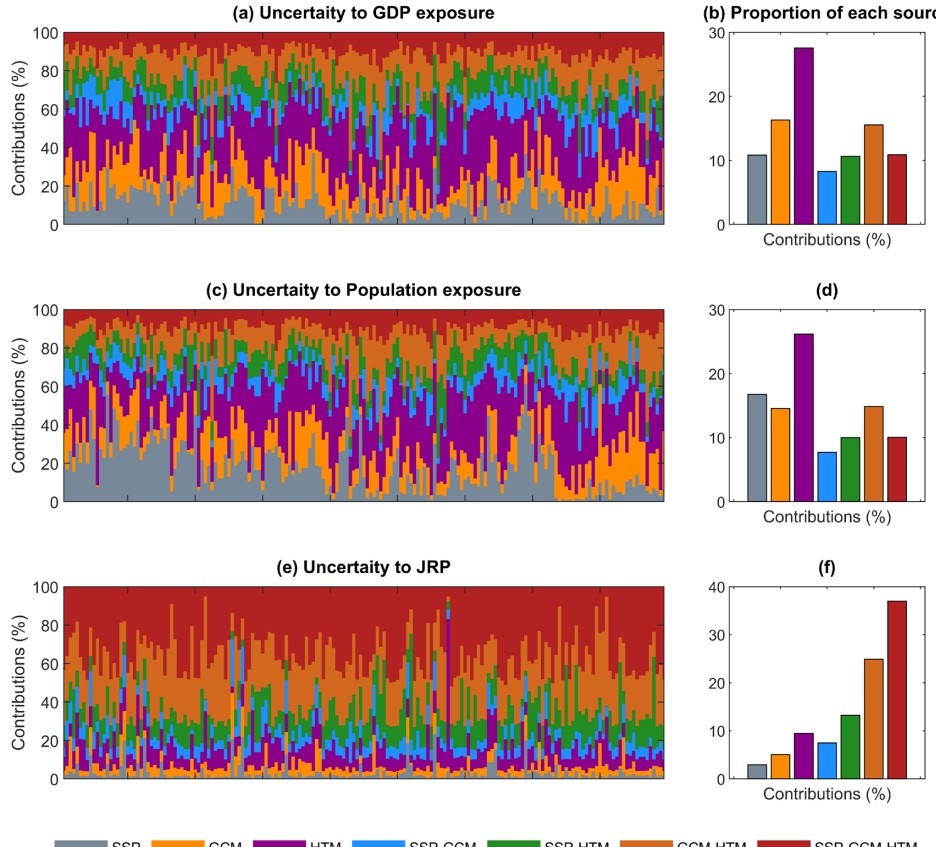

**Figure 13. The fractional uncertainty contributions of all sources to the GDP exposure, population exposure, and JRP estimate for all 179 catchments (a, c, e) and the average fractional contribution of each source (b, d, f).**

**5.2 Limitations and future work**

As hydrological drought is a complex weather-related hazard influenced by both nature and human

intervention, further work is still required to reveal the principles of drought generation. Although the hybrid

models show good performance in streamflow simulation over the selected period, the underlying uncertainty



and the coupling relationships behind interrelated variables remain unexplained in this study. Therefore, the
study of the interactions among data sources is important to reveal the drivers affecting the water cycle under
climate change. Here, only five GCM outputs and one in situ observation dataset were used to drive our HTM
models. The sparse dataset may undermine the robustness of the approach, particularly when attempting to
simulate extreme drought events (e.g., the extreme drought in the Yangtze River Basin in 2022). Although
the machine learning model show good performance herein, significantly reducing the reliance on
observational data, continuous streamflow observations are still important to improve model accuracy.
Providing a larger number of GCMs and observational data to assemble a more sophisticated model might
be an effective approach to improve the accuracy and reliability of the model. Finally, the GDP and population
projections cannot well reflect future economic development and population migration. In particular,
government interference in immigration policies is likely to lead to large uncertainties in the projections.
Therefore, considering the dynamic impact of human management on socioeconomic development is
essential for the construction of a reliable projection framework.
**5.3 Suggestions for drought mitigation in China**
In order to curb global warming and mitigate the threats by climate change, the Chinese government is
striving to reach its carbon peak before 2030, achieve carbon neutrality before 2060, and bolster efforts in
disaster reduction (Kundzewicz et al., 2019; Liu et al., 2022b). China has nonetheless experienced several
extreme drought events during the past 5 years, threatening the population's health and economic
development. (Ding and Gao, 2020; Mallapaty, 2022; Liu et al., 2022a) The Intergovernmental Panel on
Climate Change (IPCC) has emphasized that projections of future climate trends can equip policymakers
with the scientific insight needed to navigate the challenges of climate change (Pörtner et al., 2022). The
results of this study aim to alert policymakers to drought risk in Southwestern China, which is expected to
intensify with climate change. Preserving local ecological balance and employing rational use of water
resources could be the key in mitigating potential losses from extreme droughts (Sohn et al., 2016; Chang et
al., 2019). Finally, this work highlights the importance of strictly implementing carbon emission reduction
initiatives and developing prevention programs to limit potential drought losses.
**6. Conclusions**
In this study, the hybrid LSTM-constrained hydrological models show high accuracy in studied
catchments over China, demonstrating that machine learning can effectively constrain the hydrological
projections. Projected changes in 50-year bivariate drought characteristics, expressed as a JRP, indicate that
the risk of hydrological drought is likely to more than double in over 60% of catchments by the end of the
21$^{st}$ century under SSP5-85. The spatial distribution of change reveals that the catchments with severely
increased drought risk are mainly located in southwestern China. Notably, the exposure of GDP and
population varies greatly across different SSPs. The median GDP exposure under SSP5-85 is 1.5 times that



of SSP3-70, but the median population exposure is just 40% that of SSP3-70. The higher population exposure
under SSP3-70 can be attributed to rapid population growth. Finally, we find the interaction between multiple
sources of data explains more than 80% of the uncertainty in future changes in JRPs, showing the importance
of considering the relationships between model components. Our findings demonstrate that China is facing
a high risk of drought under climate change and rising pressures on population and economic growth,
emphasizing the urgency of achieving carbon neutrality goals and implementing strategies to reduce carbon
emissions.

**Data availability**



The gridded meteorological dataset for China can be obtained from http://www.cma.gov.cn. The
ISIMIP3b data can be downloaded from https://data.isimip.org. The ERA5-Land data can be
downloaded from https://www.ecmwf.int/en/era5-land. Streamflow simulations used in this study
are available at https://osf.io/fvyse/.

**Acknowledgments**


J.Y. acknowledges support from the National Natural Science Foundation of China (Grant NOs.
52009091; 52242904; 52261145744) and the Fundamental Research Funds for the Central Universities
(NO. 2042022kf1221). L.S. is supported by UKRI (MR/V022008/1). J.G. is supported by the National
Natural Science Foundation of China (NO. 52179018). This work is also supported by the
Undergraduate Training Programs for Innovation and Entrepreneurship of Wuhan University. The
numerical calculations in this paper have been performed on the supercomputing system in the
Supercomputing Center of Wuhan University.

**Competing interests**


At least one of the (co-)authors is a member of the editorial board of Hydrology and Earth System
Sciences.

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
