# Peer review of "Machine learning-constrained projection of bivariate hydrological drought magnitudes and socioeconomic risks"

_Hydrology and Earth System Sciences, 2023_

## Author Comment (AC1)

**References**

Arsenault, R., Essou, G. R., and Brissette, F. P.: Improving hydrological model simulations with combined multi-input and multimodel averaging frameworks, Journal of Hydrologic Engineering, 22, 04016066, 2017.

Cai, X., Zeng, R., Kang, W. H., Song, J., and Valocchi, A. J.: Strategic Planning for Drought Mitigation under Climate Change, Journal of Water Resources Planning and Management, 141, 04015004, https://doi.org/10.1061/(ASCE)WR.1943-5452.0000510, 2015.

Chang, J., Guo, A., Wang, Y., Ha, Y., Zhang, R., Xue, L., and Tu, Z.: Reservoir operations to mitigate drought effects with a hedging policy triggered by the drought prevention limiting water level, Water Resources Research, 55, 904–922, 2019.

Duan, Q., Sorooshian, S., and Gupta, V.: Effective and efficient global optimization for conceptual rainfall-runoff models, , 28, 1015–1031, 1992.

Hu, C., Guo, S., Xiong, L., and Peng, D.: A modified Xinanjiang model and its application in northern China, Hydrology Research, 36, 175–192, 2005.

Piao, S., Ciais, P., Huang, Y., Shen, Z., Peng, S., Li, J., Zhou, L., Liu, H., Ma, Y., Ding, Y., Friedlingstein, P., Liu, C., Tan, K., Yu, Y., Zhang, T., and Fang, J.: The impacts of climate change on water resources and agriculture in China, Nature, 467, 43–51, https://doi.org/10.1038/nature09364, 2010.

Pörtner, H.-O., Roberts, D. C., Poloczanska, E. S., Mintenbeck, K., Tignor, M., Alegría, A., Craig, M., Langsdorf, S., Löschke, S., and Möller, V.: IPCC, 2022: Summary for policymakers, 2022.

Ren-Jun, Z.: The Xinanjiang model applied in China, Journal of hydrology, 135, 371–381, 1992.

Sohn, J. A., Saha, S., and Bauhus, J.: Potential of forest thinning to mitigate drought stress: A meta-analysis, Forest Ecology and Management, 380, 261–273, https://doi.org/10.1016/j.foreco.2016.07.046, 2016.

Tian, Y., Xu, Y.-P., and Zhang, X.-J.: Assessment of Climate Change Impacts on River High Flows through Comparative Use of GR4J, HBV and Xinanjiang Models, Water Resources Management, 27, 2871–2888, 2013.

Wilhite, D. A., Svoboda, M. D., and Hayes, M. J.: Understanding the complex impacts of drought: A key to enhancing drought mitigation and preparedness, Water Resour Manage, 21, 763–774, https://doi.org/10.1007/s11269-006-9076-5, 2007.

Xiao-jun, W., Jian-yun, Z., Shahid, S., ElMahdi, A., Rui-min, H., Zhen-xin, B., and Ali, M.: Water resources management strategy for adaptation to droughts in China, Mitig Adapt Strateg Glob Change, 17, 923–937, https://doi.org/10.1007/s11027-011-9352-4, 2012.

---

## Author Response (AR1)

**Cover Letter**

February 20, 2024

Dear Editor,

We would like to thank you, the associate editor and the anonymous reviewers for constructive comments and suggestions, which have significantly improved our manuscript (**hess-2023-181**).

Climate change accelerates the water cycle, thus complicating the projection of future streamflow and hydrological droughts. Although machine learning is increasingly employed for hydrological simulations, few studies have used it to project hydrological droughts, not to mention the bivariate risks of drought duration and severity as well as their socioeconomic implications under climate change. We developed a cascade modeling chain to project future bivariate hydrological drought characteristics in 179 catchments over China, using 5 bias-corrected GCM outputs under three shared socioeconomic pathways, five hydrological models and a deep learning model. Our hybrid model also projected substantial GDP and population exposures by increasing bivariate drought risks, suggesting an urgent need to design climate mitigation strategies toward a sustainable development pathway.

In this revision, all the reviewers' concerns have been addressed. Changes made in the revised manuscript are coloured in blue. We sincerely hope you will find the revised version of the paper appropriate for publication. All authors have reviewed the paper and agree to the resubmission of the manuscript. We look forward to hearing from you.

Sincerely yours,

Jiabo Yin

Associate Professor, Wuhan University, China

Honorary Research Associate, University of Oxford, UK

**Reply to Reviewers' comments**

Legend

Reviewers' comments

Authors' responses

Direct quotes from the revised manuscript

Associate Editor:

Thank you for including a road map of your paper at the end of the introduction. Authors might also mention the section numbers where every topic is presented. I will not delay discussion any further and will start review and discussion.

**Reply:** We have rephrased this section in the revised Introduction as follows:

This study illustrated the used materials and methods in Section 2 and Section 3, respectively. We compared SRI and TWS-DSI in assessing drought conditions in Section 4.1. The contribution of meteorological factors to simulate streamflow and the calibration of hybrid terrestrial models were shown in Section 4.2. The evolution of univariate droughts was projected in Section 4.3. The bivariate droughts of future scenarios and associated socioeconomic exposures were evaluated in Section 4.4. We discussed the uncertainty of our analysis and main limitations of this study in Section 5, and finally summarized our work in Section 6.

Reviewer #1:

1. In Line 59, I'm not sure about the relationship between these two sentences. Can the author's example support the previous sentence? In my opinion, it seems like it cannot.

**Reply:** We have rephrased these sentences in the revised Introduction as follows:

China's socioeconomic development, and particularly its agricultural sector, is threatened by the rapid intensification of extreme hazards under climate change (Piao et al., 2010). Over the past years, China has been hit by severe drought events which have caused considerable damage to ecosystem productivity and socio-economic growth (Zhai and Zou, 2005; Yin et al., 2023). For instance, one extreme drought in Sichuan Province in 2022 resulted in power shortages and led to economic losses of 669 million dollars. Water shortage is also a key challenge that hinders the sustainable development of the North China Plain (Chen and Yang, 2013). Over the period of 1985-2014, drought accounted for about 19% of economic losses among all meteorological hazards (Chen and Sun, 2019). With continuing global warming, the economic losses from severe drought events might

increase by over ten billion US dollars per year by the late 21st century, underscoring the importance of projecting future droughts over China (Lu et al., 2023).

2. In the last paragraph of the Introduction section, it is recommended that the author use the past tense when describing the work done in this paper.

**Reply:** These sentences have been corrected to past tense.

2. The title of section 5.3 of the article is "Suggestions for drought mitigation in China", but I didn't seem to find any relevant suggestions based on the research results in this section (maybe I misunderstood?). In my opinion, the author presented the research background and significance of the paper to readers in section 5.2, but did not provide reasonable suggestions for reducing drought events in China. I think this is also of interest to readers, and I suggest that the author engage in a detailed discussion of this part of the content.

**Reply:** Thank you; we have provided more details about the relevant suggestions in the revised Section 5.3 as follows:

The Intergovernmental Panel on Climate Change (IPCC) has emphasized that projections of future climate trends can equip policymakers with the scientific insight needed to navigate the challenges of climate change (Pörtner et al., 2022). The results of this study aim to alert policymakers to drought risk in Southwestern China which was just hit by severe drought events and expected to significantly intensify with climate change. We strongly highlight the importance of strictly implementing carbon emission reduction initiatives and developing prevention programs to limit potential drought losses. Preserving local ecological balance and employing rational use of water resources could be the key to mitigating potential losses from extreme droughts (Sohn et al., 2016; Chang et al., 2019). Although China has constructed hydraulic structures with a total water storage capacity of over 7,064 billion m3, current irrigation facilities need to expand to mitigate the challenge of drought under climate change (Xiao-jun et al., 2012; Cai et al., 2015). In addition, it is also beneficial for policymakers that establish a drought information system to get a comprehensive collection of drought impacts from all potential sectors, which can link the government and research organizations (Wilhite et al., 2007).

3. The author is suggested to abbreviate the content in Section 5.2 and include it in the summary section after the conclusion.

**Reply:** We have revised accordingly and summarize in the revised Section 5.2 and Section 6 as follows:

The uncertainty caused by underlying surface situation and the coupling relationships behind interrelated variables remain unexplained in this study. Therefore, revealing interactions among multisource data is important to understand how the drivers affecting the water cycle under climate change. Here, only five GCM outputs and one in situ observation dataset were used to drive our HTM models. The sparse dataset may undermine the robustness of the approach. Providing a larger number of GCMs and observational data to assemble a more sophisticated model might be an effective approach to improve the accuracy and reliability. On the other hand, due to the heterogeneity of different climatic regions in China, we would like to expand hydrological models (e.g., the weather research and forecasting model hydrological modeling system, soil and water assessment tool or the hydrological modules of land surface process models) to reduce uncertainty in future research. Finally, the GDP and population projections cannot well reflect future economic development and population migration, especially the governmental interference in immigration and economic policies. It is better to consider the dynamic impact of human management on socioeconomic development, which is essential for the construction of a more reliable projection framework.

Our study is insufficient in the revelation of drought hazard drivers and needs to expand datasets and hydrological models to promote the reliability of simulation in future studies. We would also like to take governmental interference of economic and demographic policies into consideration.

4. The last sentence of the paper's conclusion, in my opinion, suggests that drought events are just one of the possible disasters that may be exacerbated by climate change, and the author did not discuss the connection between achieving carbon neutrality goals and drought events. Perhaps I did not express myself clearly, but what I meant was that there was no corresponding data in the paper to support the latter half of the sentence.

**Reply:** We have rephrased this sentence in the revised Section 6 as follows:

Our findings demonstrate that China will face higher drought risks in a warmer future, emphasizing the urgency of implementing strategies to reduce carbon emissions.

5. I noticed that the author used five hydrological models to conduct experiments, but these models all have some limitations to varying degrees. For example, how did the author consider the applicability of the XAJ hydrological model in different climatic regions, and whether the model parameters obtained using the SCE-UA algorithm are universal? Are these parameters effective if spatial heterogeneity is strictly considered? In addition, the author's introduction of these five hydrological models is too brief, and it is suggested that the author provide slightly more detail in this section. Furthermore, can the author try to use hydrological models such as WRF-Hydro, SWAT, or the hydrological modules of land surface process models to conduct experiments in this paper? In my opinion, these general hydrological models would be more convincing.

**Reply:** As the hydrological models might show uncertainties, we employed five different models as candidates to simulate streamflow. In implementing this process, we calibrate our models in each catchment. In other words, the parameters of hydrological models in different catchments are not universal. We have provided more details about the hydrological models in revised Section 2.3.1 as follows:

The XAJ (Xinanjiang) model is a hydrological model, which can usually achieve better performance in humid and semi-humid areas than in arid areas (Zhao, 1992). As the model was developed based on the underlying surface of the Yangtze River Basin in China, it is composed of a three-layer evapotranspiration module with four parameters and separates the runoff into four components (i.e., surface water, groundwater, interflow water and flow routing) (Tian et al., 2013). To date, it is widely reported that the XAJ model usually shows the best accuracy in simulating hydrological conditions in China (Hu et al., 2005). However, due to inadequacies in the simulation of arid regions, the results of the XAJ model did not be considered as the best option in northern China.

We use the SCE-UA (Shuffled Complex Evolution) approach to maximize the objective function (i.e., Kling-Gupta efficiency) to optimize these models (Duan et al., 1992). The most complete 20-year observation period is selected to calibrate five models in each watershed. To calibrate the hydrological models, a cross-validation method developed by Arsenault et al. (2017) is used for calibration, which employs the odd years of data to calibrate models, and the even years of data to validate. As catchments are located in different climatic regions, the parameters of models are calibrated for each catchment, which means that the parameters are not universal. Although uncertainties shown by hydrological models are ineradicable, the overall uncertainty is acceptable in the current scale after optimizing five hydrological models for each catchment.

We believe your suggestion of using WRF-Hydro and SWAT models is great, but it beyond the scope of this manuscript. We have discussed the future application in the revised Section 5.2 as follows:

On the other hand, due to the heterogeneity of different climatic regions in China, we would like to expand hydrological models (e.g., the weather research and forecasting model hydrological modeling system, soil and water assessment tool or the hydrological modules of land surface process models) to reduce uncertainty in future research.

Reviewer #2:

1. Text errors should be addressed:

   ● In line 384, please remove the redundant period.

   ● In line 198, is it supposed to be 'retained' instead of 'retailed'?

   ● In line 278, 'demote' should be revised to 'denote'.

**Reply:** These errors have been corrected.

2. The titles in Figure 10 are suggested to be revised, as they only include severity and duration under SSP1-26 and SSP3-70, which isn't consistent with the elaboration in section 4.3.

**Reply:** The elaboration in the manuscript is correct, and the title error of this figure have been corrected.

3. In lines 222 and 223, the number of HTMs differs from subsequent elaboration in line 284. The number of HTMs should be corrected.

**Reply:** We have corrected the number of HTMs in lines 284 to ten.

4. In lines 235 and 401, tables of drought classification and candidate distributions are suggested to be referenced from the supplement file.

**Reply:** We have provided citations in the revised Section 2.4 as follows:

The hydrological drought classification and ranges indicated by SRI are shown in Table S1.

Based on the maximum Bayesian Information Criterion (BIC), we select the best-performing marginal distributions for duration and severity from seven candidate distributions shown in Table S2, based on historical data for each catchment.

5. In Figure 6, considering the limited number of stations located in Northwestern China and the use of interpolated methods for calculating sensitivities, does it potentially affect the accuracy of the analysis? The author is suggested to elaborate on the reliability of the results in section 4.2.

**Reply:** We have provided an explanation in the revised Section 4.2 as follows:

Due to the sparse number of observation stations in Northwestern China, the reliability of the sensitivity analysis for these regions is lower than that of the dense observed areas.

6. In line 375, the author is suggested to revise this conclusion. First, it should be mentioned which type of drought is sensitive to temperature. Second, whether the feedback of drought to temperature is reliable should be discussed, as drought is affected by both hydrological and thermal factors. Univariate sensitivity isn't a powerful support under global warming.

**Reply:** We have rephrased this conclusion in the revised Section 4.2 as follows:

The temperature has a positive contribution to streamflow generation in Northeast China, suggesting a potential mitigation for the deficiency of surface flow. However, there is interactive feedback between hydrological and thermal factors that result in an inability to directly assess the impact of temperature on hydrologic droughts.

7. The author is suggested to add an explanation of which approach was used for the analysis in section 4.2. Is this analysis conducted at spatial, temporal, or spatial-temporal dimensions? More specifically, is the input data for the RF model the multi-year average of each variable from each grid (spatial), or spatial average at each timestep (temporal), or variable for each timestep for each grid (spatial-temporal)?

**Reply:** We have provided an explanation in the revised Section 4.2 as follows:

We quantified the sensitivity of seven historical mean meteorological variables (i.e., pr, ps, SH, RH, srlds, srsds, temperature) to monthly streamflow in each grid.

---

## Referee Report (RR1)

**Referee comment on the paper:**

**Machine learning-constrained projection of bivariate hydrological drought magnitudes and socioeconomic risks**

By Rutong Liu, Jiabo Yin, Louise Slater, Shengyu Kang, Yuanhang Yang, Pan Liu, Jiali Guo, Xihui Gu, Xiang Zhang, Aliaksandr Volchak

**General comments**

This paper outlines the development of a method consisting in the execution a sequence of analytical and simulation techniques, such as deep learning – long short-term memory neural networks, hybrid modelling, bivariate Copula-based analysis, machine learning and multivariate analysis of variance. After performing multiple hydrological simulations, the most likely realization approach allows to project climate change effects on the duration and severity of hydrological droughts in 179 catchments of China, while assessing their uncertainties, and the interaction and potential impact on population and gross domestic product under three shared socioeconomic pathways.

I consider that the proposed methodology is well founded and has been applied with precision, based on very complete and good quality data and observations. In my opinion, the strategy of using the joint probability density of drought characteristics to optimize the most likely realization for the selection of joint design values was particularly interesting. Similarly, the finding that simulated streamflow performance is better in the southern region of China relative to the northern region, and that this may be attributed to a greater dependence of streamflow on rainfall in South China, which is governed by a humid climate pattern, is of utmost relevance in this study. The paper is very well written, with interesting and clear tables and figures. Notwithstanding the above, minor clarifications and additions from the authors could further facilitate the good explanation offered on the conceptualization and application of the chosen cascade modelling techniques, as well as the interpretation of the results and conclusions obtained.

**Specific comments**

In **L.22** and **L.40** it is mentioned that climate change (or a warming world) accelerates the (global) water cycle, but this statement is not very precise. The hydrological cycle is a complex phenomenon, and much more elaboration would be required to generally state that, as such, it is accelerated by climate change. It seems to me that Allan et al. (2020), cited here, refer to the expectation of acceleration of global precipitation responses (as warming increases and aerosol forcing decreases), providing quotes on how the warming influence of continued rises in $CO_2$ concentration + declining aerosol cooling is expected to accelerate increases in global precipitation and its extremes as transient climate change progresses. They also point that nonlinear changes in streamflow over multidecadal timescales are expected in some regions as accelerated glacier melt is followed by declining glacier volume. However, these findings do not necessarily support the far-reaching argument for an accelerated global water cycle.

**L.51**: I would suggest reviewing the use of the adjective "uneven" to refer to the distribution of precipitation under the effects of climate change. In principle, precipitation is already an uneven phenomenon, and perhaps this term could be replaced by "rapidly changing", or another that is more precise and refers rather to some process of change than to a static characteristic.

**L.264-265**: It may be worth explaining further what is meant by selecting joint design values according to "the same frequency hypothesis" that has been applied in previous studies.

**L.480** and **L.536**: Although the word accuracy is commonly used to assess model performance in many publications, I would suggest double-checking whether it applies here. The performance of a model can be determined in terms of its efficiency (it has good predictive skills that can be tested by measures such as the Nash-Sutcliffe model efficiency coefficient), as well as by uncertainty analysis that allows the model to be characterized in terms of precision (how well each modelled value agrees with each other, i.e., width of confidence intervals constructed around modelled values) and accuracy (how well modelled values agree with "true" values, i.e., percentage of observed test values contained within certain confidence intervals built around modelled values).

**L.353-355**: Very interesting indeed is the finding that the severity of droughts measured by the TWS-DSI index is twice that of the hydrological drought, and perhaps in addition to the explanation that the TWS-DSI metric incorporates all vertical water fluxes, thus offering a comprehensive view of shifts in water scarcity, a conceptual discussion could be included around the concepts and differences between meteorological, hydrological and agricultural droughts. In this case, it should be noted that TWS-DSI does not involve aquifer recharge processes, which are fundamental to explain baseflow and, therefore, the hydrological drought in its entire extension, especially for catchments with aquifer recharge and storage capacity that exceeds several times the time step of the analysis.

**L.376-380**: How are these percentages calculated? Are the relationships of *SH*, *RH*, radiation, etc. established considering the entire tributary catchment area to the respective streamflow gauging site? Otherwise, point-to-point comparisons on the location of gauging sites would be meaningless since streamflow is a catchment-supported process. Also, what does a sensitivity rate >10% mean? Is it significant? Are the negative contributions of *RH* and shortwave radiation to streamflow significant? What do you mean by a "pronounced" negative sensitivity of shortwave radiation? If comparisons are made with catchment-support and not point-to-point, as indicated above, it does not seem to make sense that *RH* has an opposite effect on streamflow at 179 stations. Is this relationship statistically significant?

**Technical corrections**

Below I recommend technical and typographical corrections to this manuscript, and some typing suggestions.

Some acronyms or abbreviations in the document are not defined or appear for the first time without having been defined, such as: GCM (**L.27**), GDP (**L.37**), HMs (**L.117**), ML (**L.128/Fig.1**), POP (not defined), $T_{or}$ (**L.272 & L.275/Fig.3**), KGE (L.391).

Consider homogenizing/equating the use of terms such as streamflow/runoff, watershed/catchment, etc.

I would strongly advise including a table of abbreviations in the paper.

**L.98-99**: …GCMs outputs under the Coupled Model Intercomparison Project phase six (CMIP6)…

**L.100**: …to quantify the sensitivity of daily streamflow to different meteorological variables.

**L.116-117**: I would suggest including some reference for the ERA5-Land 116 dataset.

**L.128/Fig.1**: I would advise including the MANOVA analysis as a process here.

**L.131**: Perhaps you could include an introductory paragraph to section 2.1, to explain why you are calculating 2-meter relative and specific humidities.

**L.132-133**: How is Eq.(1) deriving air temperature $T$?

**L.135**: Constants $T_0$, $e_0$, $L_0$ and $R_0$ could be further explained

**L.137-138 & L.140**: Does it imply (since it is not mentioned) that $T_{2m}$, $T_2$ and $ps$ are available in ERA5?

**L.144**: … The RF model is used to calculate the sensitivity of runoff to different meteorological variables …

**L.144**: I would suggest including some reference for the RF model.

**L.153**: But is there any modeling that has been done without all the meteorological variables?

**L.171**: I do not agree with the assessment that a model containing 21 parameters is simple and efficient.

**L.177-178**: I would not consider infiltration as a type of runoff.

**L.184-185**: Could you provide more references in addition to Hu et al. (2005), or further explanation, to support the statement: "To date, it is widely reported that the XAJ model usually shows the best accuracy in simulating hydrological conditions in China"?

**L.188**: We used the SCE-UA…

**L.189-190**: "The most complete 20-year observation period is selected to calibrate five models in each watershed." At this point it might be convenient to specify the modelling time step. Is it a daily time step?

**L.211/Eq.(9)**: Aren't the subindexes *oh* and *ox* inverted in $W_{oh}$ and $W_{ox}$, in relation to the orders employed in Eqs.(5),(6)&(7)?

**L.213-214**: Are $W_{\bullet f}$, $W_{\bullet i}$, $W_{\bullet \hat{c}}$ and $W_{\bullet o}$ from Eqs(5), (6), (7)&(9) also weights?

**L.215**: In "…are the cell state of the LSTM and the hidden unit at  time $t$, respectively; $c_{t-1}$ and $hs_{t-1}$ at the former…", could you please further explain the term "hidden unit"?

**L.219-220**: If in the following statement HMs stand for Hydrological Models: "…The hydrological outputs together with other climate variables are used as inputs to feed the LSTM model (i.e., the HMs are thus constrained by the LSTM)…", then I would say that the LSTM are the ones that are being constrained by the HMs, and not the other way around.

**L.237**: What is $\overline{TWSA_y}$ and where is it used?

**L.247**: Why do you calculate two drought indexes, if Table S1 only classifies drought according to DI? If so, what is the SRI for? It is not clearly stated, but it seems like sometimes you use TWS-DSI as DI, and other times you use SRI (also not explicitly stated). Could you clarify when TWS-DSI s used and when SRI? Also, could you please explain in more detail how the maps in Figs. 4 & 5 are calculated?

**L.263**: "which contains infinite combinations of values of these two multivariate arrays of variables."

**L.275-277**: I suggest including the terms $d_T$, $s_T$, $F_S$, $F_D$ and $T_{or}$ in the legend of Fig.3 and explaining what each of them is.

**L.279**: Could you clarify what does it mean the expression "the future period" here?

**L.280**: Could you please clarify why you consider this definition of socioeconomic exposure to be "static"? (At this point you had only mentioned that it varied from 0 to 100%).

**L.286**: Please note that the conditions $T_h - Tf > 0$ and $T_h - T_f > 0$ are not mutually exclusive!

**L.287**: "…$POP$ and $GDP$ denote the population and the gross domestic product (in USD) of a given catchment in the future climate, respectively…."

**L.302**: "…is quantified by the variance of each source  to the total variance…."

**L.311-312**: "…with at least 20 years of data  were selected…"

**L.319-320**: "…As these three mason solutions are produced at different spatial resolutions, we  generated blended TWS data based on the…"

**L.324**: "…precipitation, temperature air pressure, etc. The spatial resolution of the dataset is 9 km…"

**L.328**: I assume that here $T_{dew}$ is the dew-point temperature, which in section 2.1 you first called $T_d$. Please check consistency.

**L.331-332**: "…The climate outputs of five GCMs  of the historical scenario and three SSPs (i.e., SSP1-26, SSP3-70, 331 SSP5-85) under CMIP6 are used to represent different climate scenarios…"

**L.344**: "4.1 Observed changes in SRI and TWS-DSI based drought"

**L.346**: "…employed the TWS-DSI as a supplement…."

**L.348-349**: In relation to Figs. 4 & 5, which describe drought trends based on TWS-DSI and SRI, I did not find a clear explanation in the paper on how these maps were calculated, and then several questions arise. For example, is Fig. 4 somehow calculated using Eq.(12) and how, considering that this equation is month-specific? Furthermore, it is indicated that the maps in Fig.4 correspond to the periods 2002-2022, but Fig.5 does not provide any reference time period. Could you include further description about this in the methodology section?

**L.350**: Could you explain in more detail how you concluded that drought hazards have increased in recent decades? If I understood correctly, at this point in the paper you have only provided spatial trends, not temporal trends.

**L.350-353**: How did you estimate the percentage increase in TWS-DSI droughts? Again, maybe you produced monthly maps using Eq.(12) and analyzed temporal trends for specific locations on the map? In Figs. 4 & 5 I cannot distinguish any catchment. It looks like a normal grid-based GIS analysis, rather than a catchment-based analysis. Are you using streamflow measurements to draw conclusions about hydrological droughts?

**L.364/Fig.4**: It might be convenient to explain the unit of measurement "/10 years" for drought trends, since although it is a little more intuitive in the case of frequency (count of events in a 10-year period) and duration (number of months over a 10-year period), in the case of the TWSA-DSI index this might be less obvious.

**L.381**: When you say "This" do you mean increasing or decreasing *RH*-streamflow relationships?

**L.386**: Could you refer to the different maps of Fig.6 here?

**L.389/Fig. 6**: Why do you use a thin plate smoothing spline method to interpolate your data, rather than more data-driven techniques such as directional kriging? Also, when you say point-based station data, are you referring to your hydrological stations?

**L.391/Fig. 7**: This is a very interesting figure, but it needs better explanation. For example, the legend of Fig. 7a should indicate that the regions colored according to the best performing models are the study catchments. Fig. 7b should perhaps include labels for the different categories/models, since the use of only colors is a little ambiguous.

**L.418**: Higher carbon emissions and other climate forcing factors should be listed and explained (at least briefly) in the methodology section.

**L.434-436**: A better description could be provided for terms such as IPSL_CM6A_LR. Perhaps you could also better explain in the methodological section the geo-statistics behind terms like median relative change of severity.

**L.438**: Regarding the finding of substantial spatial heterogeneity of drought across China: Are the study catchments distributed homogeneously and equally throughout the country?

Considering geospatial sampling techniques, a homogeneous density of catchments may be necessary to reach such a conclusion, in a strict sense.

**L.440**: "…intensification as a result of global warming.…"

**L.441/Fig. 10**: Color legends seem to be missing for the five GCMs.

**L.445**: "…drought severity and duration, we used a Copula…"

**L.448**: "…The medians of the projected future JRP are 38.78 years, 14.52 years and 19.24 years under…"

**L.449**: "…under SSP3-70 and SSP-5-85…"

**L.455**: The use of bivariate drought analysis can "synthesize", or rather "amplify" the individual effects of two drought characteristics?

**L.457**: Are the figures contained in Figure 12 absolute or marginal? If they are absolute, you should also present the relative change with respect to the period of reference.

**L.479**: It is very surprising to me that HTM is the main source of uncertainty, since this analysis also includes SSPs, which, being products of socioeconomic studies and models, I would think involve much higher levels of uncertainty. Does this analysis quantify uncertainties, or rather the variance explained by each of the factors?

**L.505**: Is "interference" the right term? Perhaps "intervention" would be more appropriate. In addition to that, how about other high impacting factors such as political and economic crises, changes in culture and expectations of the populations, and wars and other conflicts?

**L.545**: In line with previous comments, it really seems that your uncertainty analysis is returning rather explained variance, and not induced uncertainty.

---

## Author Response (AR2)

**Cover Letter**

May 31, 2024

Dear Editor,

We would like to thank you, the editor and the three reviewers for constructive comments and suggestions, which have significantly improved our manuscript (**hess-2023-181**).

Climate change accelerates the water cycle, thus complicating the projection of future streamflow and hydrological droughts. Although machine learning is increasingly employed for hydrological simulations, few studies have used it to project hydrological droughts, not to mention the bivariate risks of drought duration and severity as well as their socioeconomic implications under climate change. We developed a cascade modeling chain to project future bivariate hydrological drought characteristics in 179 catchments over China, using 5 bias-corrected GCM outputs under three shared socioeconomic pathways, five hydrological models and a deep learning model. Our hybrid model also projected substantial GDP and population exposures by increasing bivariate drought risks, suggesting an urgent need to design climate mitigation strategies toward a sustainable development pathway.

In this revision, all the reviewers' concerns have been addressed. Changes made in the revised manuscript are coloured in blue. We sincerely hope you will find the revised version of the paper appropriate for publication. All authors have reviewed the paper and agree to the resubmission of the manuscript. We look forward to hearing from you.

Sincerely yours,

Jiabo Yin

Associate Professor, Wuhan University, China

Honorary Research Associate, University of Oxford, UK

Editor, Journal of Environmental Management

Youth Editor, The Innovation

Email: jboyn@whu.edu.cn

**Reply to Reviewers' comments**

Legend

Reviewers' comments

Authors' responses

Direct quotes from the revised manuscript

Editor Lelys Bravo de Guenni:

Thank you to the authors for the revised version of this manuscript and to the reviewers for great comments and suggestions.

Two of the referees have proposed minor revisions which I consider would enhanced even more the quality of your work.

Reviewer 2 has made interesting comments, and suggested some clarifications on several calculations, as for example, the calculation on the sensitivity of several meteorological variables to daily streamflow in lines 376-380.

Reviewer 3 mainly suggests ideas for boosting the paper's presentation quality and writing-up clarity.

I think the reviewers have made a great job in reading the paper in detail and proving useful comments to further improve a manuscript that, not surprisingly, might achieve a good number of citations.

**Reply:** Dear Prof. Guenni, we would like to thank you for providing these helpful suggestions to improve our manuscript and proceeding the revision of our paper.

We have added explanation on the concerns from reviewer 2 in Section 4.2 as follows:

Over 30% and 38% of stations show the *SH* sensitivity rate of >10% in Western and Northeastern China respectively, indicating the dominance of *SH* in these areas.

Since a station can be attributed to catchments of different sizes, we only consider the largest catchment scales in analysis.

These negative contributions mean enhancement of these two variables will inhibit the generation of streamflow, showing the potential adverse effects of climate change on streamflow generation.

We have improved the expression according to the comments from two reviewers.

Referee #2:

I have no further comment, the manuscript can be published.

**Reply:** We appreciate your positive evaluation of our manuscript.

Reviewer #3 Gómez-Delgado, Federico:

1. In L.22 and L.40 it is mentioned that climate change (or a warming world) accelerates the (global) water cycle, but this statement is not very precise. The hydrological cycle is a complex phenomenon, and much more elaboration would be required to generally state that, as such, it is accelerated by climate change. It seems to me that Allan et al. (2020), cited here, refer to the expectation of acceleration of global precipitation responses (as warming increases and aerosol forcing decreases), providing quotes on how the warming influence of continued rises in $CO_2$ concentration + declining aerosol cooling is expected to accelerate increases in global precipitation and its extremes as transient climate change progresses. They also point that nonlinear changes in streamflow over multidecadal timescales are expected in some regions as accelerated glacier melt is followed by declining glacier volume. However, these findings do not necessarily support the far-reaching argument for an accelerated global water cycle.

**Reply:** Dear Prof. Federico, we would like to thank you for providing these insightful suggestions to improve our manuscript. We have rephrased these sentences in the revised Abstract as follows:

Climate change influences the global water cycle and alters the spatiotemporal distribution of hydrological variables, thus complicating the projection of future streamflow and hydrological droughts.

2. L.51: I would suggest reviewing the use of the adjective "uneven" to refer to the distribution of precipitation under the effects of climate change. In principle, precipitation is already an uneven phenomenon, and perhaps this term could be replaced by "rapidly changing", or another that is more precise and refers rather to some process of change than to a static characteristic.

**Reply:** We have replaced the word in the revised Introduction as follows:

The rapidly changing distribution of precipitation and other meteorological elements under climate change complicates projection of future runoff and drought.

3. L.264-265: It may be worth explaining further what is meant by selecting joint design values according to "the same frequency hypothesis" that has been applied in previous studies.

**Reply:** We have reshaped the sentence in the revised Section 2.5 as follows:

Previous studies have only selected joint design values according to the same frequency hypothesis that considering two correlated variables follow the same cumulative probability in their distributions, but this approach lacks a statistical basis and poorly describes the physical characteristics of droughts (Yin et al., 2018).

4. L.480 and L.536: Although the word accuracy is commonly used to assess model

performance in many publications, I would suggest double-checking whether it applies here. The performance of a model can be determined in terms of its efficiency (it has good predictive skills that can be tested by measures such as the Nash-Sutcliffe model efficiency coefficient), as well as by uncertainty analysis that allows the model to be characterized in terms of precision (how well each modelled value agrees with each other, i.e., width of confidence intervals constructed around modelled values) and accuracy (how well modelled values agree with "true" values, i.e., percentage of observed test values contained within certain confidence intervals built around modelled values).

**Reply:** Of course, the accuracy should indicate how well modelled values agree with "true" values, which has been quantified by variations of simulations and observations in this study. The predictive skills of models should be stated as efficiency, which has been quantified by Kling-Gupta efficiency. We have revised expression in Conclusion as follows:

In this study, the hybrid LSTM-constrained hydrological models show high efficiency in studied catchments over China, demonstrating that machine learning can effectively constrain the hydrological simulation.

5. L.353-355: Very interesting indeed is the finding that the severity of droughts measured by the TWS-DSI index is twice that of the hydrological drought, and perhaps in addition to the explanation that the TWS-DSI metric incorporates all vertical water fluxes, thus offering a comprehensive view of shifts in water scarcity, a conceptual discussion could be included around the concepts and differences between meteorological, hydrological and agricultural droughts. In this case, it should be noted that TWS-DSI does not involve aquifer recharge processes, which are fundamental to explain baseflow and, therefore, the hydrological drought in its entire extension, especially for catchments with aquifer recharge and storage capacity that exceeds several times the time step of the analysis.

**Reply:** We have added discussion in the Section 4.1 as follows:

On the other hand, TWS-DSI can difficultly represent the aquifer recharge processes, which are fundamental physical process of baseflow and the hydrological drought in its entire extension. Therefore, catchments with aquifer recharge and storage capacity will exceed several times the time step of the analysis, enlarging the severity of droughts.

6. L.376-380: How are these percentages calculated? Are the relationships of SH, RH, radiation, etc. established considering the entire tributary catchment area to the respective streamflow gauging site? Otherwise, point-to-point comparisons on the location of gauging sites would be meaningless since streamflow is a catchment-supported process. Also, what does a sensitivity rate >10% mean? Is it significant? Are

the negative contributions of RH and shortwave radiation to streamflow significant? What do you mean by a "pronounced" negative sensitivity of shortwave radiation? If comparisons are made with catchment-support and not point-to-point, as indicated above, it does not seem to make sense that RH has an opposite effect on streamflow at 179 stations. Is this relationship statistically significant?

**Reply:** These percentages are calculated by dividing stations with a sensitivity rate >10% by the number of total stations. Since a station can be attributed to catchments of different sizes, we only consider the largest catchment scales. The sensitivity rate >10% is used to describe the spatial distribution of SH in Fig. 6c, which is dominant compared with other variables. The negative contributions mean enhancement of these two variables will inhibit the generation of streamflow.

We have rephrased statement and added explanation in Section 4.2 as follows:

Over 30% and 38% of stations show the *SH* sensitivity rate of >10% in Western and Northeastern China respectively, indicating the dominance of *SH* in these areas.

Since a station can be attributed to catchments of different sizes, we only considered the largest catchment scales in analysis.

These negative contributions mean enhancement of these two variables will inhibit the generation of streamflow, showing the potential adverse effects of climate change on streamflow generation.

7. Some acronyms or abbreviations in the document are not defined or appear for the first time without having been defined, such as: GCM (L.27), GDP (L.37), HMs (L.117), ML (L.128/Fig.1), POP (not defined), Tor (L.272 & L.275/Fig.3), KGE (L.391). Consider homogenizing/equating the use of terms such as streamflow/runoff, watershed/catchment, etc. I would strongly advise including a table of abbreviations in the paper.

**Reply:** We have added a table of abbreviations in the Supplement.

Table S3. Affiliation of acronyms and their full names in this study.

| | Acronyms | Full names |
|---|---|---|
| | CMIP6 | Coupled Model Intercomparison Project phase Six |
| | SSP | Shared Socioeconomic Pathways |
| | ISIMIP3b | Intersectoral Impact Model Intercomparison Project 3b |
| **Drivers** | GCM | Global Climate Model |
| | ECMWF | European Center for Medium Weather Forecasting |
| | ERA5 | Fifth generation ECMWF Atmospheric Reanalysis of the global climate |
| **Meteorological** | *RH* | Relative Humidity |

| variables | SH | Specific Humidity |
|---|---|---|
| | $ps$ | Near surface air pressure |
| | $pr$ | Precipitation |
| | $srsds$ | Surface Downwelling Shortwave Radiation |
| | $srlds$ | Surface Downwelling Longwave Radiation |
| | $T_{2m}$ | 2-meter Temperature |
| | $T_d$ | Dew-point Temperature |
| Hydrological models | GR4J | Génie Rural à 4 paramètres Journalier |
| | HBV | Hydrologiska Byråns Vattenbalansavdelning |
| | HMETS | Hydrological Model of École de Technologie Supérieure |
| | SIMHYD | Simple lumped conceptual daily rainfall-runoff |
| | XAJ | Xinanjiang |
| Statistical & Machine learning methods | SCE-UA | Shuffled Complex Evolution |
| | BIC | Bayesian Information Criterion |
| | MANOVA | Multivariate Analysis of Variance |
| | RNN | Recurrent Neural Network |
| | LSTM | Long Short-Term Memory neural network |
| | RM | Random Forest |
| | HTM | Hybrid Terrestrial Model |
| Supporting test data | GRACE | Gravity Recovery and Climate Experiment |
| | GRACE-FO | GRACE Follow-On |
| | TWS | Terrestrial Water Storage |
| Statistical indicators | KGE | Kling-Gupta Efficiency |
| | JRP | Joint Return Period |
| Drought indicators | SRI | Standardized Runoff Index |
| | TWS-DSI | TWS based Drought Severity Index |

8. L.98-99: …GCMs outputs under the Coupled Model Intercomparison Project phase six (CMIP6)…

**Reply:** We have revised accordingly.

9. L.100: …to quantify the sensitivity of daily streamflow to different meteorological variables to daily streamflow.

**Reply:** We have revised accordingly.

10. L.116-117: I would suggest including some reference for the ERA5-Land dataset.

**Reply:** The reference of the ERA5-Land dataset has been cited in the Section 3.3.

11. L.128/Fig.1: I would advise including the MANOVA analysis as a process here.

**Reply:** We have revised the Fig.1 by adding the Multivariate analysis of variance (MANOVA) as a process as follows:

[Figure]

12. L.131: Perhaps you could include an introductory paragraph to section 2.1, to explain why you are calculating 2-meter relative and specific humidities.

**Reply:** We have added explanation in Section 2.1 as follows:

As relative humidity and specific humidity are not directly available from the ERA5-land dataset, we estimate these two variables based on the physical relationship in atmosphere.

13. L.132-133: How is Eq. (1) deriving air temperature T?

**Reply:** We derived the temperature from ERA5-Land dataset, which is mentioned in Section 3.3. The Eq. (1) is to calculate *RH* and *SH* in Eq. (2) and Eq. (3); therefore, the air temperature here indicates the 2m air temperature and the dew-point temperature.

14. L.135: Constants $T_0$, $e_0$, $L_0$ and $R_0$ could be further explained

**Reply:** We have added explanation in Section 2.1 as follows:

where $T_0$, $e_0$, $L_0$ and $R_0$ are freezing temperature in Kalvin, saturated vapor pressure under freezing temperature, latent heat of vaporization and gas constant of water vapor, with a value of 273.15 K, 611 Pa, $2.5\times10^6$ J kg$^{-1}$, 461 J kg$^{-1}$ K$^{-1}$, respectively;

15. L.137-138 & L.140: Does it imply (since it is not mentioned) that $T_{2m}$, $T_d$ and ps are available in ERA5?

**Reply:** These variables were derived from ERA5-Land dataset, which is mentioned in Section 3.3.

16. L.144: … The RF model is used to calculate the sensitivity of runoff to different meteorological variables for runoff…

**Reply:** We have revised accordingly.

17. L.144: I would suggest including some reference for the RF model.

**Reply:** We have added citations of RF model as follows:

Catani, F., Lagomarsino, D., Segoni, S., and Tofani, V.: Landslide susceptibility estimation by random forests technique: sensitivity and scaling issues, Natural Hazards and Earth System Sciences, 13, 2815–2831, https://doi.org/10.5194/nhess-13-2815-2013, 2013.

18. L.153: But is there any modeling that has been done without all the meteorological variables?

**Reply:** The selected hydrological models were driven by all the meteorological variables. There is no redundant meteorological variable excluding from the random forest model.

19. L.171: I do not agree with the assessment that a model containing 21 parameters is simple and efficient.

**Reply:** We have reshaped sentence in Section 2.3.1 as follows:

The HMETS (hydrological model of École de technologie supérieure) model contains 21 parameters and two reservoirs (i.e., the saturated and vadose zones), which is considered to efficiently implement hydrological simulation in limited scales (Martel et al., 2017).

20. L.177-178: I would not consider infiltration as a type of runoff.

**Reply:** We have revised in Section 2.3.1 as follows:

There are four types of water fluxes from different sources: impervious areas, infiltration, interflow, and groundwater storage (Chiew et al., 2002).

21. L.184-185: Could you provide more references in addition to Hu et al. (2005), or further explanation, to support the statement: "To date, it is widely reported that the XAJ model usually shows the best accuracy in simulating hydrological conditions in China"?

**Reply:** We have reshaped sentence in Section 2.3.1 as follows:

To date, it is widely reported that the XAJ model usually shows a great performance in simulating hydrological conditions in China.

We have also added citations as follows:

Jiang, T., Chen, Y. D., Xu, C., Chen, X., Chen, X., and Singh, V. P.: Comparison of hydrological impacts of climate change simulated by six hydrological models in the Dongjiang Basin, South China, Journal of Hydrology, 336, 316–333, https://doi.org/10.1016/j.jhydrol.2007.01.010, 2007.

22. L.188: We used the SCE-UA…

**Reply:** We have revised accordingly.

23. L.189-190: "The most complete 20-year observation period is selected to calibrate five models in each watershed." At this point it might be convenient to specify the modelling time step. Is it a daily time step?

**Reply:** We have added information of time step in Section 2.3.1 as follows:

The most complete 20-year observation period is selected to calibrate five models in each watershed by a daily time step.

24. L.211/Eq.(9): Aren't the subindexes $oh$ and $ox$ inverted in $W_{oh}$ and $W_{ox}$, in relation to the orders employed in Eqs.(5),(6)&(7)?

**Reply:** We have revised these inverted subindexes accordingly.

25. L.213-214: Are $W_{\bullet f}$, $W_{\bullet i}$, $W_{\bullet \tilde{c}}$ and $W_{\bullet o}$ from Eqs(5), (6), (7)&(9) also weights?

**Reply:** All $W$ with any subindexes are weights of corresponding gates in Eq (5), (6), (7) & (9). We have added explanation in Section 2.3.2 as follows:

$W_{\bullet}$ are the weights, where $W_i$, $W_{\tilde{c}}$, $W_f$ and $W_o$ are the weights of each gate, $W_{x\bullet}$ are the weights of each gate at time $t$, $W_{h\bullet}$ are the weights of each gate at the former time $t-1$;

26. L.215: In "…are the cell state of the LSTM and the hidden unit at  time $t$, respectively; $ct-1$ and $hst-1$ at the former…", could you please further explain the term "hidden unit"?

**Reply:** We have revised accordingly.

27. L.219-220: If in the following statement HMs stand for Hydrological Models: "…The hydrological outputs together with other climate variables are used as inputs to feed the LSTM model (i.e., the HMs are thus constrained by the LSTM)…", then I

would say that the LSTM are the ones that are being constrained by the HMs, and not the other way around.

**Reply:** We have revised in Section 2.3.2 as follows:

i.e., the LSTM is thus constrained by the HMs

28. L.237: What is $\overline{TWSA_y}$ and where is it used?

**Reply:** It should be $\overline{TWS_y}$ which is mentioned in Eq (12). We have revised it accordingly.

29. L.247: Why do you calculate two drought indexes, if Table S1 only classifies drought according to DI? If so, what is the SRI for? It is not clearly stated, but it seems like sometimes you use TWS-DSI as DI, and other times you use SRI (also not explicitly stated). Could you clarify when TWS-DSI s used and when SRI? Also, could you please explain in more detail how the maps in Figs. 4 & 5 are calculated?

**Reply:** The drought index (DI) includes TWSA-DSI and SRI. Therefore, two indexes are simultaneously classified by Table S1. We have reshaped caption of Table S1 in Supplement as follows:

Classification of drought and threshold values of the drought events. Two drought indexes, TWSA-DSI and SRI, both follow this classification.

We have added details to explain Figs 4 and 5 in Section 4.1 as follows:

Based on linear regression and least square method, trends in drought characteristics (i.e., frequency, duration and severity) are estimated by using the GRACE/GRACE-FO dataset and observed runoff across China.

30. L.263: "which contains infinite combinations of values of these two multivariate arrays of variables."

**Reply:** We have revised accordingly.

31. L.275-277: I suggest including the terms dT, sT, FS, FD and Tor in the legend of Fig.3 and explaining what each of them is.

**Reply:** We have added explanation at the caption of Fig.3 in Section 2.5 as follows:

$d_T$ and $s_T$ are marginal distribution quantiles for a given probability level T; $F_S$ and $F_D$ are cumulative probability density of duration and severity, respectively. $T_{or}$ is a given probability level under the OR case.

32. L.279: Could you clarify what does it mean the expression "the future period" here?

**Reply:** We have added specific description in Section 2.5 as follows:

The future socioeconomic exposure after 2020s has directly been defined as ranging from 0 to 100% (Gu et al., 2020a), but dynamically shifting climate risks cannot be represented under this definition, without considering fluctuation in the frequency of hazards.

33. L.280: Could you please clarify why you consider this definition of socioeconomic exposure to be "static"? (At this point you had only mentioned that it varied from 0 to 100%).

**Reply:** We have rephrased the sentence in Section 2.5 as follows:

The future socioeconomic exposure after 2020s has directly been defined as ranging from 0 to 100% (Gu et al., 2020a), but dynamically shifting climate risks cannot be represented under this definition, without considering fluctuation in the frequency of hazards.

34. L.286: Please note that the conditions Th − Tf > 0 and Th − Tf > 0 are not mutually exclusive!

**Reply:** We have revised the incorrect symbol in Section 2.5 as follows:

$I(\cdot)$ denotes the controlling function, which is 1 when $T_h - T_f < 0$, or is 0 when $T_h - T_f \geq 0$ is recorded;

35. L.287: "…POP and (GDP) denotes the population and the gross domestic product (in USD) (GDP) of a given catchment in the future climate, respectively…."

**Reply:** We have revised accordingly.

36. L.302: "…is quantified by the variance of each source by to the total variance…."

**Reply:** We have revised accordingly.

37. L.311-312: "…with at least 20 years of data are were selected…"

**Reply:** We have revised accordingly.

38. L.319-320: "…As these three mason solutions are produced at different spatial resolutions, we produce generated blended TWS data based on the…"

**Reply:** We have revised accordingly.

39. L.324: "…precipitation, temperature, and air pressure, etc. The spatial resolution of

the dataset is 9 km…"

**Reply:** We have revised accordingly.

40. L.328: I assume that here $T_{dew}$ is the dew-point temperature, which in section 2.1 you first called $T_d$. Please check consistency.

**Reply:** It should be $T_d$. We have revised in whole paper.

41. L.331-332: "…The climate outputs of five GCMs  of the historical scenario and three SSPs (i.e., SSP1-26, SSP3-70, 331 SSP5-85) under CMIP6 are used to represent different climate scenarios…"

**Reply:** We have revised accordingly.

42. L.344: "4.1 Observed changes in SRI and TWS-DSI based drought"

**Reply:** We have revised accordingly.

43. L.346: "…employed the TWS-DSI as a supplement…."

**Reply:** We have revised accordingly.

44. L.348-349: In relation to Figs. 4 & 5, which describe drought trends based on TWS-DSI and SRI, I did not find a clear explanation in the paper on how these maps were calculated, and then several questions arise. For example, is Fig. 4 somehow calculated using Eq.(12) and how, considering that this equation is month-specific? Furthermore, it is indicated that the maps in Fig.4 correspond to the periods 2002-2022, but Fig.5 does not provide any reference time period. Could you include further description about this in the methodology section?

**Reply:** We have added explanation in Section 4.1 as follows:

Based on linear regression and least square method, Trends in drought characteristics (i.e., frequency, duration and severity) are estimated by using the GRACE/GRACE-FO dataset and observed runoff across China.

We have added time period at the caption of Fig. 5 in Section 4.1 as follows:

Trends in drought frequency, duration and severity from 2002 to 2022 over China. (c), the index of severity is based on the SRI statistic (Eq. 13).

45. L.350: Could you explain in more detail how you concluded that drought hazards have increased in recent decades? If I understood correctly, at this point in the paper

you have only provided spatial trends, not temporal trends.

**Reply:** We have rephrased this sentence in Section 4.1 as follows:

Overall, the two indexes show similar trends in most catchments, suggesting that drought hazards have increased during 2002-2022.

46. L.350-353: How did you estimate the percentage increase in TWS-DSI droughts? Again, maybe you produced monthly maps using Eq.(12) and analyzed temporal trends for specific locations on the map? In Figs. 4 & 5 I cannot distinguish any catchment. It looks like a normal grid-based GIS analysis, rather than a catchment-based analysis. Are you using streamflow measurements to draw conclusions about hydrological droughts?

**Reply:** It should be grid here. The percentage is calculated by gridded results. We have reshaped in Section 4.1 as follows:

TWS-DSI droughts have increased in 54% of grids, which are mainly located in the Qinghai-Tibet Plateau, the North China Plain and the northwestern Xinjiang Province.

47. L.364/Fig.4: It might be convenient to explain the unit of measurement "/10 years" for drought trends, since although it is a little more intuitive in the case of frequency (count of events in a 10-year period) and duration (number of months over a 10-year period), in the case of the TWSA-DSI index this might be less obvious.

**Reply:** The TWSA-DSI represents the severity of drought. Although the severity has less obvious trends, we also analyzed it in the unit of "/10 years" consisting with other drought characteristics for showing changes of drought condition.

48. L.381: When you say "This" do you mean increasing or decreasing RH-streamflow relationships?

**Reply:** Yes, **"This"** refers to the finding about RH-streamflow relationships mentioned above.

49. L.386: Could you refer to the different maps of Fig.6 here?

**Reply:** We have added this information in Section 4.2 as follows:
(Fig. 6i and 6f)

50. L.389/Fig. 6: Why do you use a thin plate smoothing spline method to interpolate your data, rather than more data-driven techniques such as directional kriging? Also, when you say point-based station data, are you referring to your hydrological stations?

**Reply:** The directional kriging method sounds a great alternative. We will consider this in further studies.

The point-based station data refers to observation dataset mentioned in Section 3.1. It includes hydrological variables.

51. L.391/Fig. 7: This is a very interesting figure, but it needs better explanation. For example, the legend of Fig. 7a should indicate that the regions colored according to the best performing models are the study catchments. Fig. 7b should perhaps include labels for the different categories/models, since the use of only colors is a little ambiguous.

**Reply:** We have added explanation in the caption of Fig.7 as follows:

(a), The best-performing model with the highest KGE value. The catchments are colored according to the best performing models.

52. L.418: Higher carbon emissions and other climate forcing factors should be listed and explained (at least briefly) in the methodology section.

**Reply:** We have added explanation in Section 3.4 as follows:

Generally, the SSP5-85 configured the highest carbon emission and human interference with the natural environment. The SSP3-70 and the SSP1-26 have progressively conservative changes to represent climate change resulting from different levels of human activity.

53. L.434-436: A better description could be provided for terms such as IPSL_CM6A_LR. Perhaps you could also better explain in the methodological section the geo-statistics behind terms like median relative change of severity.

**Reply:** We have reshaped sentence in Section 4.3 as follows:

The median relative change of severity based on the IPSL-CM6A-LR under SSP3-70 are 30%, and 22% of catchments have a relative change over 200%, representing the most severe case of drought evolution.

54. L.438: Regarding the finding of substantial spatial heterogeneity of drought across China: Are the study catchments distributed homogeneously and equally throughout the country? Considering geospatial sampling techniques, a homogeneous density of catchments may be necessary to reach such a conclusion, in a strict sense.

**Reply:** The studied catchments cover all the nine major basins within China and basically satisfy a homogeneous spatial distribution. The density of studied catchments is also consistent with the spatial density distribution of river networks in China.

We have added explanation in Section 5.2 as follows:

Although the catchments gathered in this study cover nine major watersheds in China, there is still

a requirement for streamflow data with a more uniform spatial density. Considering geospatial sampling techniques, a homogeneous density of catchments is significant to reveal the spatial distribution of drought.

55. L.440: "…intensification as a result of global warming.…"

**Reply:** We have revised accordingly.

56. L.441/Fig. 10: Color legends seem to be missing for the five GCMs.

**Reply:** We have revised Fig.10 as follows:

[Figure]

57. L.445: "…drought severity and duration, we used a Copula…"

**Reply:** We have revised accordingly.

58. L.448: "…The medians of the projected future JRP are 38.78 years, 14.52 years and 19.24 years under…"

**Reply:** We have revised accordingly.

59. L.449: "…under SSP3-70 and SSP-5-85…"

**Reply:** We have revised accordingly.

60. L.455: The use of bivariate drought analysis can "synthesize", or rather "amplify" the individual effects of two drought characteristics?

**Reply:** We have revised accordingly.

61. L.457: Are the figures contained in Figure 12 absolute or marginal? If they are absolute, you should also present the relative change with respect to the period of reference.

**Reply:** The Fig.12 shows the exposure of GDP and population in the 2071-2100 time period, which is defined by changes in drought frequency and demographic and economic fundamental of catchments stated in Section 2.5. So, the exposure has considered the relative change with respect to the period of reference.

62. L.479: It is very surprising to me that HTM is the main source of uncertainty, since this analysis also includes SSPs, which, being products of socioeconomic studies and models, I would think involve much higher levels of uncertainty. Does this analysis quantify uncertainties, or rather the variance explained by each of the factors?

**Reply:** The uncertainty analysis in this study, as explained in Section 2, based on the MNOVA and quantified the contribution of data from different sources to the variance of the results. In other words, it can be considered as the uncertainty of each component in the cascade model chain.

63. L.505: Is "interference" the right term? Perhaps "intervention" would be more appropriate. In addition to that, how about other high impacting factors such as political and economic crises, changes in culture and expectations of the populations, and wars and other conflicts?

**Reply:** The "intervention" is more appropriate, we have revised accordingly.   The listed social impacting factors absolutely have a high potential influence, but beyond the attention of this study. We would like to consider these aspects in future studies.

64. L.545: In line with previous comments, it really seems that your uncertainty analysis is returning rather explained variance, and not induced uncertainty.

**Reply:** The uncertainty analysis in this study, as explained in Section 2, based on the

MNOVA and quantified the contribution of data from different sources to the variance of the results. In other words, it can be considered as the uncertainty of each component in the cascade model chain.

Reviewer #4:

1. Starting at the highest level, as most diagrams concern China, it feels like this country domain should be mentioned in the title. Then, in the Abstract, please make it much clearer what some of the terminology refers to. For instance, the extensive use of "bivariate" – is this referring to the two variables (i.e. "bi") of drought duration and severity? Or maybe water storage and runoff? Although points like this are made clearer in the paper, the Abstract should be as complete as possible for people to understand the analysis. I am sure this can be achieved without making the Abstract excessively long.

**Reply:** We have rephrased the title of this study as follows:

Machine learning-constrained projection of bivariate hydrological drought magnitudes and socioeconomic risks over China

We have rephrased sentence in Abstract as follows:

Although machine learning is increasingly employed for hydrological simulations, few studies have used it to project hydrological droughts, not to mention the bivariate drought risks, referring to drought duration and severity, as well as their socioeconomic effects under climate change.

2. The paper has a particularly large number of acronyms. A reader trying to understand the manuscript will be quickly attracted to the schematic of Figure 1, which sets out the methodological components. However, it is then necessary to work back through the manuscript to discover all of the acronyms. Would the authors like to consider a table? In the Table there would be different sets of rows explaining all acronyms of (1) drivers (ECMWF, CMIP6, SSP), (2) hydrological models (SIMHYD etc), (3) key meteorological variables (RH, srlds, srsds), (4) statistical / AI methods used (LTSM….), (5) Supporting test data (GRACE…), (6) statistics of performance (e.g. KGE). I reckon there are at least 30 acronyms in this paper, and a single point where all are listed would be extremely helpful to the reader.

**Reply:** We have added a table of acronyms in Supplement as follows:

Table S1. Affiliation of acronyms and their full names in this study.

| | Acronyms | Full names |
|---|---|---|
| **Drivers** | CMIP6 | Coupled Model Intercomparison Project phase Six |

| | SSP | Shared Socioeconomic Pathways |
|---|---|---|
| | ISIMIP3b | Intersectoral Impact Model Intercomparison Project 3b |
| | GCM | Global Climate Model |
| | ECMWF | European Center for Medium Weather Forecasting |
| | ERA5 | Fifth generation ECMWF Atmospheric Reanalysis of the global climate |
| **Meteorological variables** | $RH$ | Relative Humidity |
| | $SH$ | Specific Humidity |
| | $ps$ | Near surface air pressure |
| | $pr$ | Precipitation |
| | $srsds$ | Surface Downwelling Shortwave Radiation |
| | $srlds$ | Surface Downwelling Longwave Radiation |
| | $T_{2m}$ | 2-meter Temperature |
| | $T_d$ | Dew-point Temperature |
| **Hydrological models** | GR4J | Génie Rural à 4 paramètres Journalier |
| | HBV | Hydrologiska Byråns Vattenbalansavdelning |
| | HMETS | Hydrological Model of École de Technologie Supérieure |
| | SIMHYD | Simple lumped conceptual daily rainfall-runoff |
| | XAJ | Xinanjiang |
| **Statistical & Machine learning methods** | SCE-UA | Shuffled Complex Evolution |
| | BIC | Bayesian Information Criterion |
| | MANOVA | Multivariate Analysis of Variance |
| | RNN | Recurrent Neural Network |
| | LSTM | Long Short-Term Memory neural network |
| | RM | Random Forest |
| | HTM | Hybrid Terrestrial Model |
| **Supporting test data** | GRACE | Gravity Recovery and Climate Experiment |
| | GRACE-FO | GRACE Follow-On |
| | TWS | Terrestrial Water Storage |
| **Statistical indicators** | KGE | Kling-Gupta Efficiency |
| | JRP | Joint Return Period |
| **Drought indicators** | SRI | Standardized Runoff Index |
| | TWS-DSI | TWS based Drought Severity Index |

3. Please make the captions complete, especially as these days, people often extract single diagrams from paper to give talks. For instance, under Figure 5, at the minimum, please state the period over which the data applies. Maybe cite back to Eqn (13) for the SRI statistic. Remove any ambiguity, e.g. that SRI statistic only applies to panel (c). (for instance, write:. "..and severity over China. The index of severity (panel (c)) is based on the SRI statistic (Eqn 13)".

**Reply:** We have revised the caption of Fig.5 as follows:

Figure 5. Trends in drought frequency, duration and severity from 2002 to 2022 over China. (c), the index of severity is based on the SRI statistic (Eq. 13).

4. Check in Figure 1 that the mention of CMIP6 is in the correct place. As it stands, being to the left of the diagram (which is more about the contemporary period), it gives the impression CMIP6 drivers may be somehow bias corrected for their projections of the historical period? i.e. part of the data assimilation. Is this the intended meaning?

**Reply:** The GCM outputs in CMIP6 has a time span from 1850 to 2100, including both the historical period (1985-2014) and the future period (2030), which is mentioned in Section 3.4. The biased corrected historical data is used to drive the models, which can be treated as a part of the data assimilation.

5. Please provide a little more background detail on the Copulus method, even if only a more definitive sentence at the beginning of Section 2.5. Please make clearer how this statistic is used in future projections. Is it to interpret combined fitted hydrological models with climate drivers – i.e. interpret future droughts only? Or is the statistic used more deeply, linking drivers with severity and duration – and then only using the ESM-based drivers. (In other words, it is an additional predictor to using conceptual hydrological models).

**Reply:** We have added explanation of Copula functions in Section 2.5 as follows:

To integrate the assessment of drought change arising from the duration and severity under climate change, we employed a Copula framework by constructing joint probability distribution of two variables.

6. Please make sure "headline" findings jump out of the paper. The most important feature of the manuscript is the Abstract sentence "By the late 21st Century, bivariate drought risk is projected to double over 60% of catchments". This summary needs to really jump out in the manuscript, and the reader taken to the key plot that illustrates this. For example, thinking of a policymaker who might not be interested in all of the details, but recognises the importance of raised drought risk. And again, even here in the Abstract, please try and help the reader. Would it be better to write "…bivariate

drought risk (which is a merged statistic capturing both drought duration and intensity)….” Also, to avoid any ambiguity, state what SSP scenario this refers to. Please check throughout the paper there are no points where misunderstanding could easily occur, and that can be resolved with a little more clarity and detail.

**Reply:** We have reshaped findings in Abstract as follows:

By the late 21st century, bivariate drought risk is projected to double over 60% of catchments mainly located in Southwest China under SSP5-85, which shows the increase of drought duration and severity.